# Olfactory specificity regulates lipid metabolism through neuroendocrine signaling in *Caenorhabditis elegans*

Ayse Sena Mutlu [1,2✉], Shihong Max Gao[2], Haining Zhang [1] & Meng C. Wang [1,2,3,4✉]

Olfactory and metabolic dysfunctions are intertwined phenomena associated with obesity and neurodegenerative diseases; yet how mechanistically olfaction regulates metabolic homeostasis remains unclear. Specificity of olfactory perception integrates diverse environmental odors and olfactory neurons expressing different receptors. Here, we report that specific but not all olfactory neurons actively regulate fat metabolism without affecting eating behaviors in *Caenorhabditis elegans*, and identified specific odors that reduce fat mobilization via inhibiting these neurons. Optogenetic activation or inhibition of the responsible olfactory neural circuit promotes the loss or gain of fat storage, respectively. Furthermore, we discovered that FLP-1 neuropeptide released from this olfactory neural circuit signals through peripheral NPR-4/neuropeptide receptor, SGK-1/serum- and glucocorticoid-inducible kinase, and specific isoforms of DAF-16/FOXO transcription factor to regulate fat storage. Our work reveals molecular mechanisms underlying olfactory regulation of fat metabolism, and suggests the association between olfactory perception specificity of each individual and his/her susceptibility to the development of obesity.

[1] Huffington Center on Aging, Baylor College of Medicine, Houston, TX 77030, USA. [2] Graduate Program in Developmental Biology, Baylor College of Medicine, Houston, TX 77030, USA. [3] Department of Molecular and Human Genetics, Baylor College of Medicine, Houston, TX 77030, USA. [4] Howard Hughes Medical Institute, Baylor College of Medicine, Houston, TX 77030, USA. ✉email: ozseker@bcm.edu; wmeng@bcm.edu

Metabolic homeostasis is fundamental for organism health, and its misregulation contributes to obesity, a worldwide epidemic in our current society. Understanding why some individuals are more susceptible to obesity has been difficult, involving a number of factors from both environmental and genetic sources. In nature, organisms constantly sense and adapt to environmental changes, and sensory systems are essential to process these environmental signals and modulate physiological responses accordingly. One of the key sensory modalities is olfactory perception, and its association with fat metabolism has been observed, but the underlying mechanism remains unknown. In both obese patients and animals, dysregulation of peripheral metabolic factors including insulin-like growth factor 1 (IGF-1) and leptin affects olfactory sensitivity[1,2]; and reducing IGF-1 receptor expression in the olfactory epithelium increases fat mass[3]. On the other hand, genetic or surgical ablation of olfactory systems modulates the sensitivity to high-fat induced obesity[3,4]. These studies imply a bi-directional association between obesity and olfactory distortion, but the results from these studies are contradictory, given that the gross removal of olfactory systems[3] and the enhancement of olfactory sensitivity[4] can both prevent high-fat induced obesity. These conflicting findings under obese conditions raise the questions, how the sense of smell regulates fat metabolism under physiological conditions, and how the specificity of olfactory perception plays a role in this regulation.

Olfactory perception is a highly regulated, complicated process, whose specificity is based on both environmental odorant signals and olfactory neurons presenting different olfactory receptors. There is a diversity of odorant molecules in the environment that can be selectively detected by specific olfactory neurons. Olfactory information is then relayed from olfactory neurons to interneurons, where it will be interpreted to command other neurons and peripheral tissues[5]. Perception of the identity and intensity of odorant molecules depends on the specificity of olfactory receptors. Interestingly, olfactory receptor polymorphisms have been associated with the onset and severity of human obesity[6]. However, whether and how a specific environmental odorant cue and its responding olfactory neurons can directly regulate peripheral fat metabolism remains unclear.

The nematode Caenorhabditis elegans is an attractive model to address this question under physiological conditions. Owing to their transparent body and recent technical development on chemical imaging of lipid molecules, fat storage dynamics can be tracked at the organism level with high temporal and spatial resolution[7]. The complete neural web in C. elegans also facilitates the discovery of neuronal regulation in fat metabolism[8]. Gustatory nutrient-sensing ADF and ASI neurons, oxygen-sensing URX neurons, and pheromone-sensing ADL neurons have been linked to fat storage in C. elegans[9–12]. However, even in C. elegans, whether and how olfactory perception plays an active role in regulating fat metabolism remains unknown.

C. elegans process olfactory signals with specificity at the levels of both environmental odorant cues and olfactory neurons, and carry 3 pairs of olfactory neurons, AWA, AWB, and AWC that detect a variety of different volatile odors[13]. Certain odors selectively activate and/or inhibit a single olfactory neuron, while some odors can stimulate a subset of neurons. In bilaterally asymmetric AWC neurons that express different receptors, the specificity of olfactory sensation is observed even between the right and left pair, where one AWC neuron responds to the odor 2-butanone, and 2,3-pentanedione targets the other[14]. This mechanism of neuronal asymmetry not only expands the diversity of olfactory neurons with distinct specificity, but also creates a circuit suited for signal integration via a diverse set of inputs to different downstream interneurons[15].

In the present study, we discovered that 2-butanone and its responding AWC olfactory neuron act through a selective neural circuit and downstream neuroendocrine pathway to directly regulate peripheral fat metabolism without affecting eating behaviors. Through genetic and optogenetic approaches, we also demonstrated that only one of asymmetric AWC neurons is crucial for this regulation, supporting the high selectivity of the olfactory sense in fine-tuning metabolic physiology.

## Results

**A guanylyl cyclase regulates fat metabolism.** To study neurons and neuroendocrine factors that regulate lipid metabolic homeostasis, we examined fat content levels in a variety of chemosensory neuronal mutants. These mutants have chemosensory defects due to different functional alterations in signal transduction, neuropeptide secretion, or sensory cilia development and differentiation[13] (Supplementary Fig. 1a). To analyze their fat content, we employed stimulated Raman scattering (SRS) microscopy, which is a chemical imaging technique for direct, label-free quantification of lipid molecules in living cells and organisms[16]. Among the 30 mutants that we screened, we identified that the mutants of daf-11, which encodes a conserved transmembrane guanylyl cyclase, show a fat storage increase (Supplementary Fig. 1a). Transmembrane guanylyl cyclases catalyze the conversion of GTP to cGMP, a secondary messenger that is widely used in sensory signal transduction across different species[17]. daf-11(m47) carries a nonsense mutation leading to deletion of the entire guanylyl cyclase catalytic domain, which is likely a null mutation[18]; while daf-11(ks67) carries a missense mutation targeting a key residue in the catalytic domain, which is likely a hypomorph[18]. We found that these two different loss-of-function alleles of daf-11 both cause fat storage increase, but the level of the increase is higher in daf-11(m47) than that in daf-11(ks67) (Fig. 1a, b). The fat storage increase caused by the daf-11 mutants predominantly occurs in the intestine, one of several major fat storage sites in C. elegans[7] (Fig. 1a), and is also confirmed by biochemical assays measuring total triglyceride levels in whole worms (Fig. 1b).

We next compared feeding behaviors and physical activities between the daf-11 mutants and wild-type controls in the first three days of adulthood, including pharyngeal pumping rate, defecation cycle duration and speed of active locomotion, and found no significant differences (Fig. 1c–e). Additionally, long-term trajectory analysis measures the proportion of different behavioral states and indicates that the daf-11 mutants exhibit no overall differences of physically active status in comparison to wild-type worms (Supplementary Fig. 2a), but spend more time in dwelling than in roaming states (Supplementary Fig. 2b). To date, it is unclear whether an increased roaming behavior is related to increased fat accumulation. Given that the che-2(e1033) mutants with increased roaming behavior[19] shows no induction of fat storage (Supplementary Fig. 1a), there might not be a direct correlation. However, we cannot exclude the possibility that the magnitude of roaming changes might be important for causing changes in fat storage and different locomotion behaviors may exhibit different demands on fat metabolism. Taking together, these results suggest that the fat storage increase in the daf-11 mutants is not due to a change in feeding behaviors, and unlikely a result of increased dwelling movements.

At the cellular level, fat storage is a highly dynamic process that is balanced between lipid synthesis and catabolism in lipid droplets. In order to track cellular lipid flux, we implemented isotope-labeling coupled SRS microscopic imaging and analyzed the rates of lipid synthesis and catabolism specifically in fat storage tissues[20]. Upon supplementing worms with deuterium-labeled oleic acid

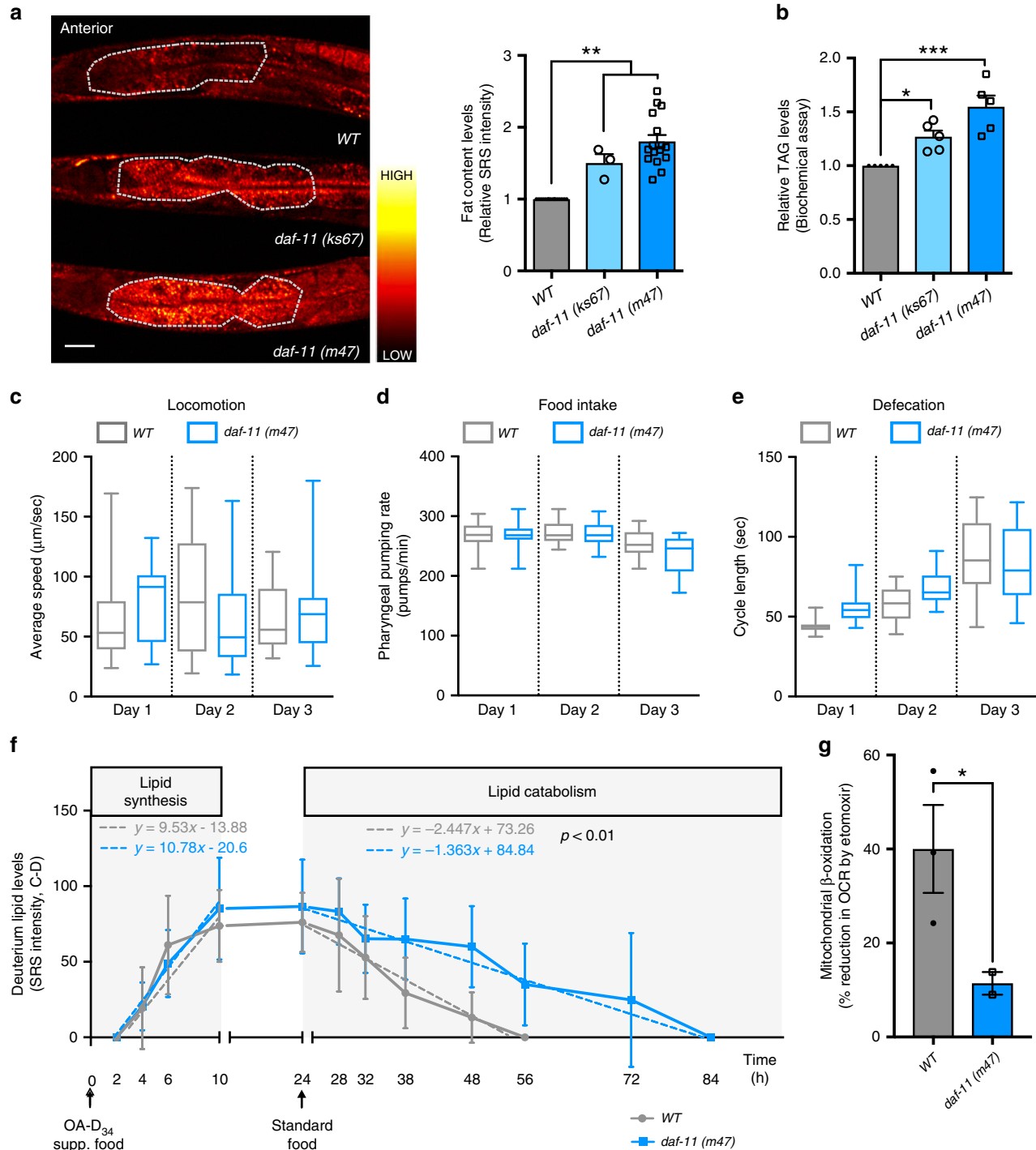

(OA-D$_{34}$), we recorded the accumulation of newly synthesized deuterium-labeled neutral lipids in intestinal lipid droplets over 24 h (Fig. 1f and Supplementary Fig. 2c). We found that the *daf-11* mutant and wild-type worms show a similar accumulation rate, suggesting that the *daf-11* mutants do not increase fatty-acid absorption or lipid synthesis. After this pulse with OA-D$_{34}$, we then transferred the deuterium-labeled worms to unlabeled plates for tracking the decay of neutral lipids from intestinal lipid droplets over the next 60 h (Fig. 1f and Supplementary Fig. 2c). We found that the *daf-11* mutants have a decreased rate of lipid catabolism (Fig. 1f and Supplementary Fig. 2c–e). We also measured oxygen consumption rates in the presence and absence of etomoxir, a

mitochondrial β-oxidation inhibitor, in order to determine the percentage of oxygen consumption due to fatty-acid oxidation. We found that mitochondrial β-oxidation is decreased in the *daf-11* mutants compared to wild-type controls (Fig. 1g). We also noted that despite the reduction in lipid catabolism, the total oxygen consumption rate is not decreased in the *daf-11* mutants (Supplementary Fig. 2f), which suggests no reduction in total energy expenditure. Together, these results reveal that the guanylyl cyclase DAF-11 directly regulates fat metabolism in peripheral fat storage tissues without changing food intake, and its mutation shifts the balance away from fat oxidation without decreasing energy expenditure.

**Fig. 1 Neuronal guanylate cyclase regulates peripheral fat mobilization. a** SRS microscopy images and quantification of fat content levels in the anterior intestine of 3-day-old adults show increased fat storage in the guanylate cyclase *daf-11* mutants, when compared to wild-type worms (WT). Data are mean ± s.e.m., **$P < 0.01$ by one-way ANOVA with Dunnett's test, scale bar = 40 μm, dashed lines indicate quantified intestine areas, yellow pixels indicate higher SRS signals. **b** Triglyceride levels measured using colorimetric enzyme assays are increased in the *daf-11* mutants, when compared to WT. Data are mean ± s.d. of five biological replicates with ~5000 worms for each genotype in each replicate, *$P < 0.05$, ***$P < 0.001$ by one-way ANOVA with Dunnett's test. **c–e** *daf-11* mutants do not show any significant changes in their locomotion (**c**), or in feeding (**d**) and defecation rates (**e**) when compared to WT. Statistical analysis with one-way ANOVA with Sidak's multiple comparison test. The boxes span the interquartile range, median is marked by the line and whiskers indicate the minimum and the maximum measurements. **f** Labeling/chasing assays with deuterium-labeled oleic acid (OA-D$_{34}$) determine rates of lipid synthesis and catabolism. Signals derived from deuterium-labeled lipids in intestinal lipid droplets were quantified at indicated time points using SRS microscopy. *daf-11* mutants have no changes in the rate of lipid synthesis, but have a significant reduction in the rate of lipid catabolism ($P < 0.01$ by linear regression analysis). Data are mean ± s.d, dashed lines represent the trendlines and their corresponding equations are indicated. SRS intensity curve represents replicate #1 (Replicate #2 is presented in Supplementary Fig. 2c). **g** Reduction in oxygen consumption rate is measured after addition of etomoxir, that blocks mitochondrial β-oxidation, using Seahorse extracellular flux analyzer. *daf-11* mutants have reduced levels of mitochondrial β-oxidation compared to wild-type worms. Data are mean ± s.e.m. of three independent biological replicates with at least ten microplate wells containing about 25 worms, *$P < 0.05$ by Student's two-tailed *t*-test. For **a** and **c–f** numbers of animals used are listed in Supplementary Data 1. For **a–g** source data are provided as a Source Data file.

**Guanylyl cyclase acts in olfactory neurons to regulate fat**. DAF-11 expresses in a group of head neurons[17], and it is a key regulator of dauer formation that is an alternative developmental stage for animals to survive unfavorable conditions such as food scarcity and overcrowding[21]. We thus investigated where DAF-11 functions to specifically modulate lipid homeostasis, and whether its roles in regulating lipid homeostasis and dauer formation are coupled. We first confirmed that *daf-11* is expressed in seven pairs of head sensory neurons: dye-filling (DiI) positive ASK, ADL, ASI, AWB, ASH, ASJ neurons and DiI negative AWC olfactory neurons (Fig. 2a). We then utilized mosaic analysis to investigate which neuron(s) is specifically involved in the lipid metabolic regulation. In *C. elegans*, extrachromosomal transgenic arrays tend to undergo spontaneous loss during meiosis and mitosis, leading to mosaic expression of transgenes among different individuals. Employing the *raxEx144[Pdaf-11::daf-11::sl2::RFP]* transgenic array along with DiO lipophilic dye that marks amphid sensory neurons with green color in the *daf-11* mutant background, we found that mosaic expression of *daf-11* in AWC neurons consistently rescues the fat storage increase (Fig. 2b and Supplementary Fig. 1b). Next, we confirmed that restoration of *daf-11* expression only in AWC neurons using the *ceh-36* short promoter is sufficient to rescue the fat storage increase in the *daf-11* mutants (Fig. 2c and Supplementary Fig. 6a). These results suggest that neural cGMP signaling specifically acts in AWC olfactory neurons to regulate peripheral fat metabolism.

It was previously known that inactivation of *daf-11* leads to constitutive dauer formation (Daf-c) during development[21]. Dauer entry and exit are tightly controlled by sensory neurons, and the activity of ASJ neurons is important for the dauer phenotype of the *daf-11* mutants[22]. We found that specific restoration of *daf-11* expression in ASJ neurons has no effect on the fat phenotype of the *daf-11* mutants (Fig. 2d and Supplementary Fig. 6b), although it rescues the Daf-c phenotype (Table 1), Thus, ASJ sensory neurons play a negligible role in regulating fat metabolism. Conversely, specific restoration of *daf-11* expression in AWC neurons fully rescues the high-fat phenotype (Fig. 2c and Supplementary Fig. 6a), but has no effects on the dauer formation in the *daf-11* mutants (Table 1). Together, these findings reveal that sensory neural circuits that regulate dauer formation and peripheral fat metabolism are distinct and independent from each other, and more importantly suggest that AWC olfactory neural activity is specifically responsible for the fat storage changes in the periphery.

**Olfactory specificity fine-tunes fat metabolism**. Interestingly, AWC are key olfactory neurons in *C. elegans* with bilateral asymmetry by expressing different receptors[23,24]. The action of

*daf-11* in AWC neurons thus indicates the importance and specificity of olfaction in regulating fat metabolism. To directly assess the effect of AWC olfactory neurons, we first examined fat storage in worms where AWC olfactory neurons are genetically ablated by targeted expression of split caspases[25]. We found that the removal of AWC olfactory neurons results in a fat storage increase in the intestine (Fig. 3a and Supplementary Fig. 6c), suggesting its active role in regulating peripheral fat metabolism cell non-autonomously.

The two AWC olfactory neurons are asymmetric with one AWC$^{ON}$ and one AWC$^{OFF}$. The AWC$^{ON}$ neuron expresses the *str-2* receptor and detects the odorant 2-butanone, whereas the AWC$^{OFF}$ neuron expresses the *srsx-3* receptor and senses the odorant 2,3-pentanedione[23]. AWC asymmetry is established stochastically in late embryogenesis via a gap junction neural network and a calcium-activated MAP kinase cascade[26]. After AWC asymmetry is established via transient signaling during embryogenesis, one of the most important pathways that maintain AWC$^{ON}$ and AWC$^{OFF}$ identities throughout the life of the animal is the cGMP pathway[27]. The *daf-11* mutants with diminished cGMP levels fail to maintain the AWC asymmetry and have two AWC$^{OFF}$ neurons[23,28]. We thus hypothesized that AWC neurons may employ their asymmetry to exert distinct effects on fat metabolism. To test this hypothesis, we examined fat storage in the *nsy-5* mutant with no AWC$^{ON}$ but two AWC$^{OFF}$ neurons and found increased levels in the intestine (Fig. 3a and Supplementary Fig. 6c). In contrast, we discovered that the *nsy-1* mutants with no AWC$^{OFF}$ but two AWC$^{ON}$ show decreased fat storage levels (Fig. 3a and Supplementary Fig. 6c). Together, these results reveal that AWC$^{ON}$ and AWC$^{OFF}$ exert different effects in regulating peripheral fat levels (Fig. 3b).

Moreover, we found that AWC neurons specifically regulate fat metabolism, with no effects on dauer formation. Neither the genetic ablation of AWC olfactory neurons nor the defective AWC asymmetry by *nsy-5* or *nsy-1* mutations causes changes in dauer formation at either low (15 °C) or high (25 °C) temperature (Table 1).

**Environmental odors dynamically regulate fat mobilization**. As olfactory neurons, AWC can be inhibited in the continuous presence of odors and get activated by odor removal[29]. Importantly, different odors can distinctly target AWC$^{ON}$ and AWC$^{OFF}$ neurons[14] (Fig. 3b). Intrigued by the discovery of AWC olfactory neurons in coordinating lipid homeostasis, we asked whether environmental odors could directly modulate fat metabolism. In order to examine the direct effect of a specific odor on lipid catabolism, we employed an odor exposure assay where we

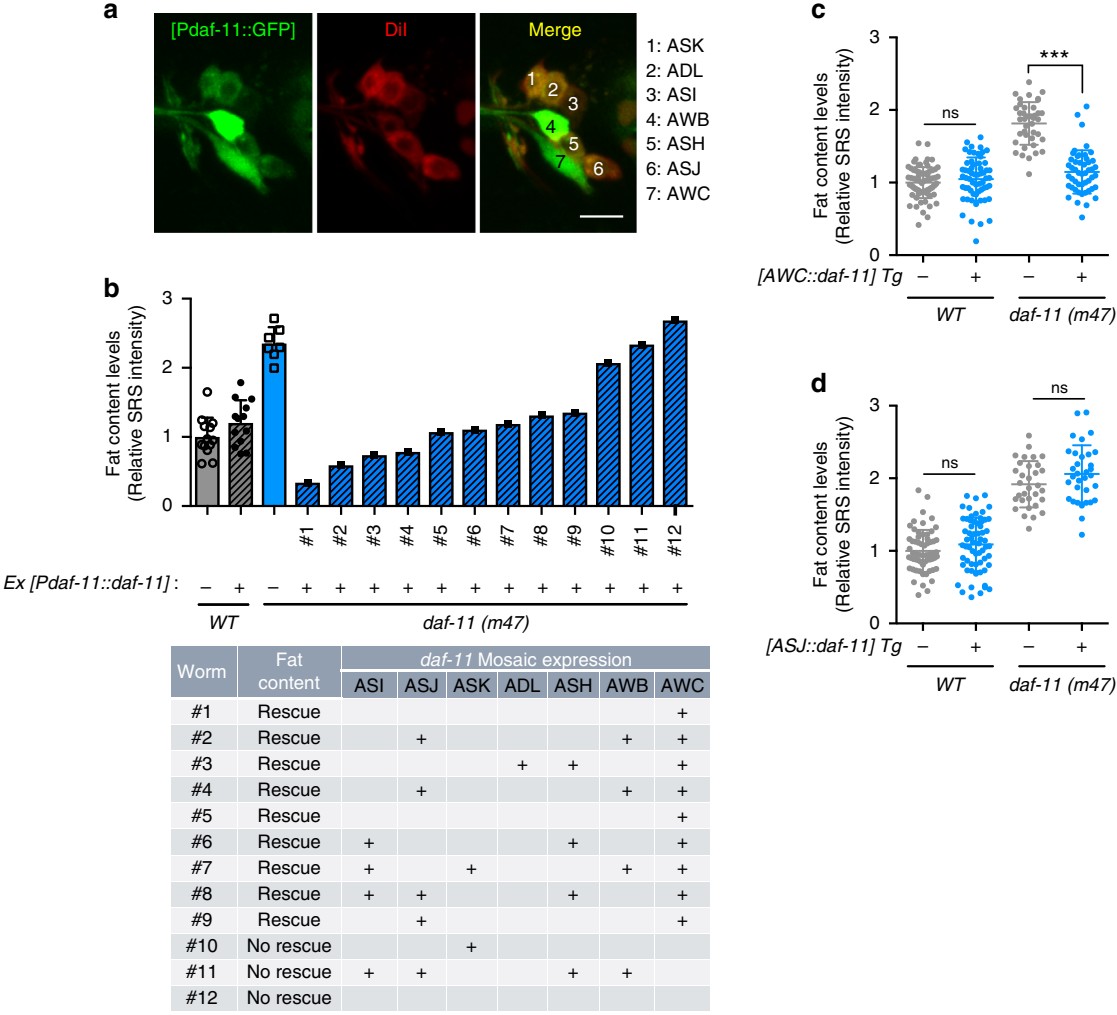

**Fig. 2 Guanylate cyclase acts in olfactory neurons to regulate fat metabolism. a** The *[daf-11::GFP]* integrated transgenic array shows the expression of *daf-11* in seven pairs of head neurons: ASK, ADL, ASI, AWB, ASH, ASJ, and AWC. Scale bar = 10 μm. **b** Mosaic analysis using the *[daf-11::RFP]* extrachromosomal transgenic array discovers that *daf-11* expression in AWC olfactory neurons sufficiently rescues the increased fat storage in the *daf-11* mutants. Individual worms that are mosaic for the transgenic array expression, are numbered 1–12 in the bar chart and in the table presenting the expression pattern. Error bars represent s.d. Individual worm images are in Supplementary Fig. 1b. **c** Using the *ceh-36* promoter, specific restoration of *daf-11* only in AWC neurons is sufficient to suppress the increased fat storage in the *daf-11* mutants. Representative SRS images are in Supplementary Fig. 6a. **d** Using the *ssu-1* promoter, restoration of *daf-11* expression specifically in ASJ neurons does not suppress the fat storage increase in the *daf-11* mutants. Representative SRS images are in Supplementary Fig. 6b. For **c** and **d**, data are mean ± s.d., ***$P < 0.001$, n.s. not significant by two-way ANOVA with Sidak's multiple comparison test. For **c–d**, numbers of animals used are listed in Supplementary Data 1. For **b–d**, source data are provided as a Source Data file.

remove animals from food to cease lipid synthesis and promote lipid mobilization. This experimental design can also eliminate the odorant interference that might stem from the food. We found that the exposure to 2-butanone that blocks the activity of AWC$^{ON}$ neurons leads to an increase in fat storage levels (Fig. 3c and Supplementary Fig. 6d), whereas exposure to other odors, including the 2,3-pentanedione that blocks AWC$^{OFF}$ neurons[14], isoamyl alcohol and benzaldehyde that are sensed by both AWC neurons[30] does not result in significant changes (Fig. 3c and Supplementary Fig. 6d). Moreover, odors that are sensed by AWB olfactory neurons, including 1-octanol and 2-nonanone[30], have no effects on fat storage levels (Fig. 3c and Supplementary Fig. 6d). Interestingly, we also found that compared to animals exposed to vehicle controls, animals exposed to 2-butanone show increased fat storage levels rapidly within 4 h (Fig. 3d), and upon removal of the odor, the increased fat storage levels are rapidly recovered (Fig. 3d). Furthermore, this increase is due to a

blockage of lipid mobilization by 2-butanone (Supplementary Fig. 3a). We further confirmed that the genetic ablation of AWC olfactory neurons suppresses the fat storage increase induced by 2-butanone exposure (Fig. 3e and Supplementary Fig. 6e). Together, these results reveal a dynamic and reversible regulation of fat metabolism by specific environmental odors, and also suggest that this selectivity is mediated not only by distinctive types of olfactory neurons, but also by the same type of olfactory neurons that carry different receptors.

To further confirm the specificity of two asymmetric AWC neurons in regulating fat metabolism, we used the *str-2* and the *srsx-3* promoter to express Channelrhodopsin in AWC$^{ON}$ and AWC$^{OFF}$ neuron, respectively, and employed optogenetic manipulation to specifically activate AWC$^{ON}$ or AWC$^{OFF}$ neuron. We found that the optogenetic activation of AWC$^{ON}$ neuron is sufficient to decrease fat storage levels, but the optogenetic activation of AWC$^{OFF}$ neuron has no effects (Fig. 3f and Supplementary Fig. 6f).

**Table 1 Dauer formation and lipid metabolism are regulated via distinct neural circuits and DAF-16/FOXO isoforms.**

| Genotype | % Dauer Formation | | Fisher's exact test, compared to *WT* | | Fisher's exact test, compared to *daf-11* | |
|---|---|---|---|---|---|---|
| | 15 °C | 25 °C | 15 °C | 25 °C | 15 °C | 25 °C |
| *WT (N2)* | 0 | 0 | – | – | $P < 0.001$ | $P < 0.001$ |
| *daf-11(m47)* | 8.9 | 71.3 | $P < 0.001$ | $P < 0.001$ | – | – |
| *daf-11; [AWC::daf-11]* | 20.7 | 62.5 | $P < 0.001$ | $P < 0.001$ | $P = 0.028$ | $P = 0.296$ |
| *daf-11; [ASJ::daf-11]* | 7.1 | 18.7 | $P < 0.001$ | $P < 0.001$ | $P = 0.795$ | $P < 0.001$ |
| *oyIs85 (AWC ablation)* | 0 | 0 | n.s. | n.s. | $P < 0.001$ | $P < 0.001$ |
| *nsy-5(ky634) (AWC$^{OFF/OFF}$)* | 0 | 0 | n.s. | n.s. | $P < 0.001$ | $P < 0.001$ |
| *nsy-1(ok593) (AWC$^{ON/ON}$)* | 0 | 0 | n.s. | n.s. | $P < 0.001$ | $P < 0.001$ |
| *daf-16(mgDf50)* | 0 | 0 | n.s. | n.s. | $P < 0.001$ | $P < 0.001$ |
| *daf-11(m47); daf-16(mgDf50)* | 1.4 | 1.0 | n.s. | n.s. | $P < 0.001$ | $P < 0.001$ |
| *daf-16; Is[daf-16a]* | 0 | 0 | n.s. | n.s. | $P < 0.001$ | $P < 0.001$ |
| *daf-16; Is[daf-16b]* | 0 | 0 | n.s. | n.s. | $P < 0.001$ | $P < 0.001$ |
| *daf-16; Is[daf-16d/f]* | 0 | 0 | n.s. | n.s. | $P < 0.001$ | $P < 0.001$ |
| *daf-11; daf-16; Is[daf-16a]* | 1.8 | 68.1 | n.s. | $P < 0.001$ | $P < 0.001$ | n.s. |
| *daf-11; daf-16; Is[daf-16b]* | 0 | 0 | n.s. | n.s. | $P < 0.001$ | $P < 0.001$ |
| *daf-11; daf-16; Is[daf-16d/f]* | 1.5 | 5.8 | n.s. | $P < 0.001$ | $P < 0.001$ | $P < 0.001$ |

The *daf-11* mutants are dauer-formation constitutive (Daf-c) at 25 °C during the development. AWC-specific restoration of *daf-11* expression has no effects on dauer formation, whereas ASJ-specific restoration suppresses the Daf-c phenotype of the *daf-11* mutants. Additionally, the worms that lack AWC completely (AWC ablation) or lost AWC asymmetry (AWC$^{OFF/OFF}$ or AWC$^{ON/ON}$) do not show any dauer formation defects. *daf-16* is required for the Daf-c phenotype of the daf-11 mutants. The *daf-16a*, but not the *daf-16d/f* isoform is sufficient to restore the dauer formation in the *daf-11*; *daf-16* double mutants. *P*-values are calculated by Fisher's exact test. Source data are provided as a Source Data file.

Upon discovering the specific effect of AWC$^{ON}$ neuron on peripheral fat metabolism, we next searched for downstream neural circuits and neuroendocrine mechanisms.

**Specific olfactory neural circuit signals to fat metabolism.** AWC neurons synapse onto different interneurons. Among them, we discovered that AIY interneurons specifically relay the olfactory inputs from AWC to peripheral fat storage. AIY interneurons are inhibited by the activity of AWC olfactory neurons[29], and the glutamate gated chloride channel, GLC-3 expressed in AIY is required for this postsynaptic inhibition[29]. In addition, the transcription factor *ttx-3* is required for AIY differentiation[31]. We found that *ttx-3* inactivation suppresses the increased fat storage induced by 2-butanone exposure (Fig. 3e and Supplementary Fig. 6e), as well as in the *nsy-5* (Fig. 3g and Supplementary Fig. 6g) and the *daf-11* mutants (Supplementary Fig. 3b) that have no AWC$^{ON}$ neuron. Furthermore, the genetic ablation of AIY interneurons by targeted expression of split caspases causes suppression similar to the *ttx-3* inactivation, and sufficiently decreases fat storage in the *daf-11* mutants that have two AWC$^{OFF}$ (Supplementary Fig. 3c). Conversely, the deletion of *glc-3*, which releases the inhibition of AIY interneurons by AWC olfactory neurons, leads to a fat storage increase in the intestine (Fig. 3g and Supplementary Fig. 6g). We also expressed the Beggitoa-photoactivated adenylyl cyclase (bPAC) under the control of the AIY-specific *ttx-3* promoter[31,32], and confirmed that the bPAC-mediated optogenetic activation of AIY interneurons is sufficient to increase fat storage (Fig. 3h and Supplementary Fig. 6h). Together, these data reveal that a specific olfactory neural circuit, including AWC$^{ON}$ olfactory neurons and AIY interneurons, regulates peripheral fat metabolism cell nonautonomously.

**FOXO mediates olfactory regulation of fat metabolism.** Next, we investigated the downstream effectors that are required for the peripheral fat storage increase caused by olfactory alterations. The nuclear hormone receptor, DAF-12 functions downstream of *daf-11* to control dauer formation[33], and can influence fat storage[34]. However, we found that the fat storage increase in the *daf-11* mutants does not require *daf-12* (Fig. 4a and Supplementary

Fig. 6i), suggesting that AWC neuronal regulation of peripheral fat metabolism is independent of DAF-12.

Insulin signaling is a crucial endocrine mechanism when considering lipid metabolic regulation in response to environmental cues[35]. Mutations of the *daf-2* insulin/IGF-1 receptor lead to increased fat storage in *C. elegans*, which depends on the FOXO transcription factor, DAF-16[35] (Supplementary Fig. 4a). We found that the inactivation of *daf-11* further enhances the fat storage increase conferred by the *daf-2* mutation (Fig. 4b and Supplementary Fig. 6j), but the inactivation of *daf-16* fully suppresses the fat storage increase in the *daf-11* mutants (Fig. 4c and Supplementary Fig. 7a). These results suggest that *daf-2* and *daf-11* may act independently to regulate fat storage but converge on *daf-16*. *daf-2* null mutants are lethal and the *daf-2(e1370)* mutant allele used here is a hypomorph. Thus, it is also possible that *daf-2* and *daf-11* may act in an overlapping pathway, and the *daf-11* inactivation enhances fat storage increase in the *daf-2* hypomorphic mutant through further reducing insulin signaling. Importantly, *daf-16* inactivation fully suppresses the fat storage increase in the *nsy-5* mutants with no AWC$^{ON}$ olfactory neurons and in the *glc-3* mutants with activated AIY interneurons (Fig. 4c and Supplementary Fig. 7a). We also detected the induction of *sod-3*, a well-characterized target gene of DAF-16, in the *daf-11, nsy-5* and *glc-3* mutants, supporting an enhanced FOXO transcriptional activity (Supplementary Fig. 4b). Together, these results suggest that the olfactory regulation of peripheral fat metabolism may be acting independently of insulin signaling, but requires the FOXO transcription factor.

The specific activity of FOXO/DAF-16 relies on its distinct isoforms and tissue distribution[36]. We discovered that the *daf-16d/f*, but not *daf-16a* or *daf-16b* isoform, sufficiently restores the fat storage increase in the *daf-11;daf-16* double mutants (Fig. 4d and Supplementary Fig. 7b), suggesting the specific involvement of the *daf-16d/f* isoform in mediating the fat storage regulation by AWC neurons. Since the *daf-16d/f* isoform is expressed in both neurons and the intestine[36], we further analyzed its functional tissues in this regulation. We found that the intestine-specific expression of the *daf-16d/f* isoform is sufficient to restore the fat storage increase in the *daf-11; daf-16* double mutants, whereas the neuron-specific expression has no such effect (Fig. 4d and Supplementary Fig. 7b). Collectively, these results show that

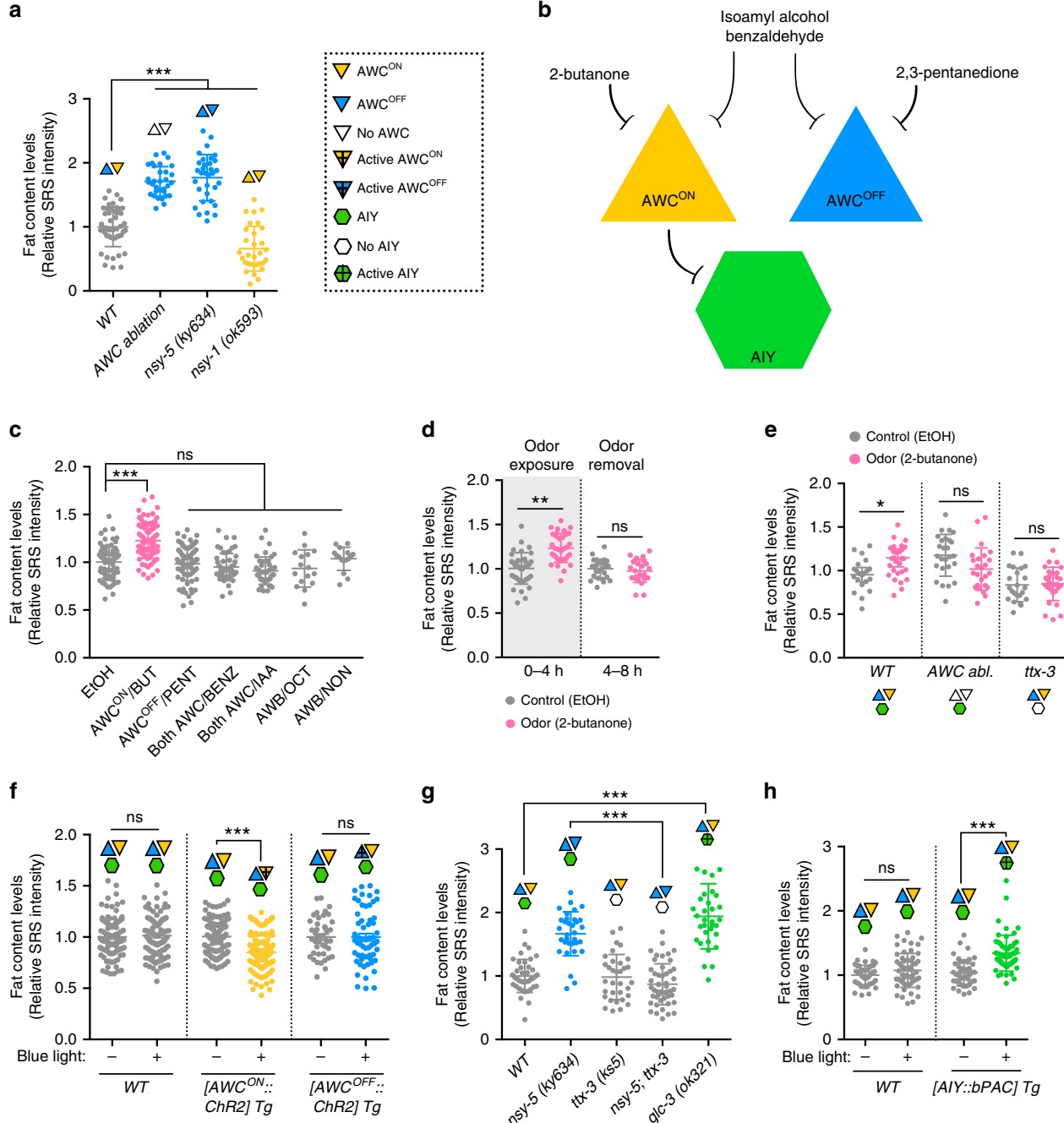

**Fig. 3 Specific odors and responding olfactory neural circuit regulate fat metabolism. a** Genetic ablation of AWC neurons increases fat storage, while the loss of the asymmetric AWC$^{OFF}$ (nsy-1 = AWC$^{ON/ON}$) or AWC$^{ON}$ (nsy-5 = AWC$^{OFF/OFF}$) neuron leads to reciprocal changes, decreasing or increasing fat levels, respectively. Representative SRS images are in Supplementary Fig. 6c. **b** Scheme of the odorants and their responding neural circuit with asymmetric AWC olfactory neurons and AIY interneurons. **c** Exposing WT to 2-butanone that inhibits AWC$^{ON}$ neurons is sufficient to increase fat levels. Other odors that target AWC$^{OFF}$, both AWC, or AWB neurons, do not change fat storage. Worms are exposed to odorants in the absence of food for 4 h. BUT 2-butanone, PENT 2,3-pentanedione, BENZ benzaldehyde, IAA isoamyl alcohol, OCT 1-octanol, NON 2-nonanone. Representative SRS images are in Supplementary Fig. 6d. **d** Exposure of WT to 2-butanone increases fat storage within 4 h. Upon odor removal, the fat levels are restored back to normal within 4 h. **e** Lack of AWC (AWC abl.) or AIY (ttx-3 mutation) suppresses the increased fat storage conferred by 2-butanone. EtOH, negative control. Representative SRS images are in Supplementary Fig. 6e. **f** ChR2-mediated optogenetic activation of AWC$^{ON}$ is sufficient to decrease fat storage levels, but optogenetic stimulation of AWC$^{OFF}$ does not affect fat levels. 1 h blue light stimulation and 1 h in dark in the absence of food followed by SRS imaging. Representative SRS images are in Supplementary Fig. 6f. **g** Lack of AIY suppresses the increased fat storage in the nsy-5 mutants. Hyperactivity of AIY (glc-3 mutation) recapitulates the fat storage increase in the nsy-5 mutants. Representative SRS images are in Supplementary Fig. 6g. **h** bPAC-mediated optogenetic activation of AIY is sufficient to increase fat storage. Representative SRS images are in Supplementary Fig. 6h. For **a** and **c** data are mean ± s.d., *P < 0.05, **P < 0.01, ***P < 0.001, n.s. not significant by one-way ANOVA with Dunnett's test and for **d**–**h** by two-way ANOVA with Sidak's test. Numbers of animals used are listed in Supplementary Data 1. For **a** and **c**–**h**, source data are provided as a Source Data file.

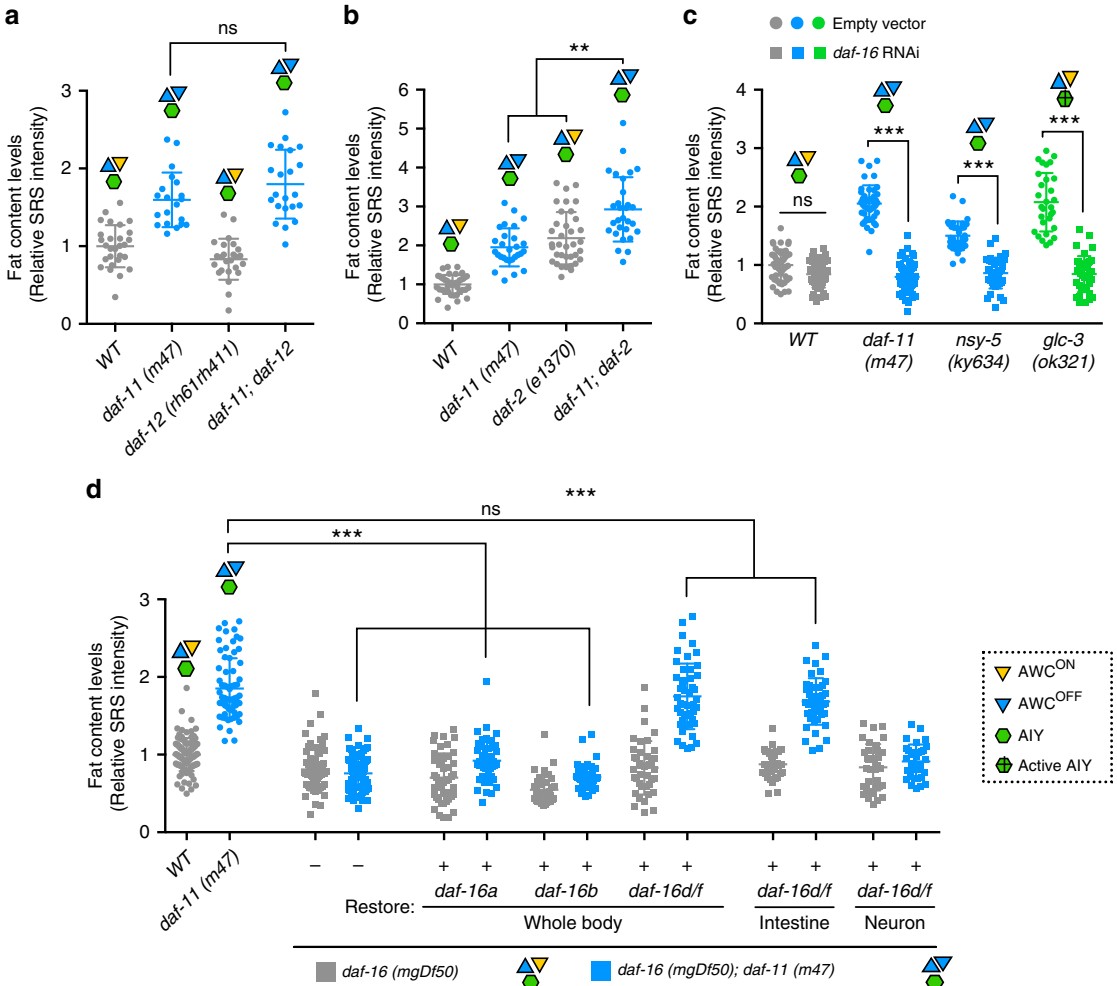

**Fig. 4 Specific FOXO isoforms mediate the olfactory regulation of fat metabolism. a** Inactivation of the nuclear hormone receptor DAF-12 does not suppress the increased fat storage in the *daf-11* mutants with only AWC^OFF neurons. Representative SRS images are in Supplementary Fig. 6i. **b** The *daf-2* mutation leads to fat storage increase, and further enhances the fat storage increase in the *daf-11* mutants. Representative SRS images are in Supplementary Fig. 6j. **c** DAF-16 is required for the increased fat storage in the *daf-11* and *nsy-5* mutants, that lack AWC^ON neurons, and in the *glc-3* mutants with activated AIY neurons. Representative SRS images are in Supplementary Fig. 7a. **d** The *daf-16* deletion fully suppresses the increased fat storage in the *daf-11* mutants, which can be rescued by restoration of the *daf-16d/f*, but not the *daf-16a* or *daf-16b* isoform. Intestine-only but not neuron-only restoration of the *daf-16d/f* isoform sufficiently rescues the fat storage increase in the *daf-11; daf-16* double mutants. Representative SRS images are in Supplementary Fig. 7b. For **a**, **b**, **c** data are mean ± s.d., **P < 0.01, ***P < 0.001, n.s. not significant by two-way ANOVA with Sidak's multiple comparison test. For **d**, data are mean ± s.d., ***P < 0.001, n.s. not significant by one-way ANOVA with Tukey's multiple comparison test. Numbers of animals used are listed in Supplementary Data 1. For **a–d** source data are provided as a Source Data file.

downstream of olfactory signaling, specific isoforms of the FOXO transcription factor act in the fat storage tissue to regulate fat metabolism cell-autonomously.

FOXO/DAF-16 is also required for the dauer phenotype of the *daf-11* mutants[21]. Considering that DAF-11 acts in distinct sensory neurons to regulate dauer formation and fat metabolism separately (Fig. 2c, d and Table 1), we questioned whether this functional separation is also mediated by different DAF-16 isoforms. We found that the *daf-16d/f* isoform, which restores fat levels in the *daf-11; daf-16* double mutants (Fig. 4d and Supplementary Fig. 7b), does not restore the Daf-c phenotype (Table 1). On the other hand, the *daf-16a* isoform is specific for the Daf-c (Table 1), but not for the fat storage phenotype of the *daf-11* mutants (Fig. 4d and Supplementary Fig. 7b). Therefore, distinct neural circuits can regulate fat metabolism vs. dauer development in parallel by acting through different isoforms of downstream transcription factors. Both the *daf-11* and the *daf-16* mutants have a reduced brood size. Distinct from its suppressing

effect on the fat storage increase, *daf-16* inactivation does not suppress the brood size reduction in the *daf-11* mutants (Supplementary Fig. 4c). Thus, the fecundity and the fat storage phenotypes of the *daf-11* mutants are regulated via distinctive mechanisms, suggesting that the fat storage increase is not caused by the defect in fecundity.

**SGK-1 regulates fat metabolism in response to olfaction.** We further searched for the factors activating FOXO/DAF-16 in response to olfactory signaling. The transcriptional activity of FOXO/DAF-16 can be regulated via two different phosphorylation mechanisms: inhibition and nuclear exclusion by the AKT complex[37], and activation by other kinases including the AMP activated protein kinase homolog, AAK-2[38], c-Jun N-terminal kinase homolog, JNK-1[39], and serum and glucocorticoid-inducible kinase homolog, SGK-1[40]. We found that unlike in the *daf-2* mutants, DAF-16 nuclear distribution is not enhanced

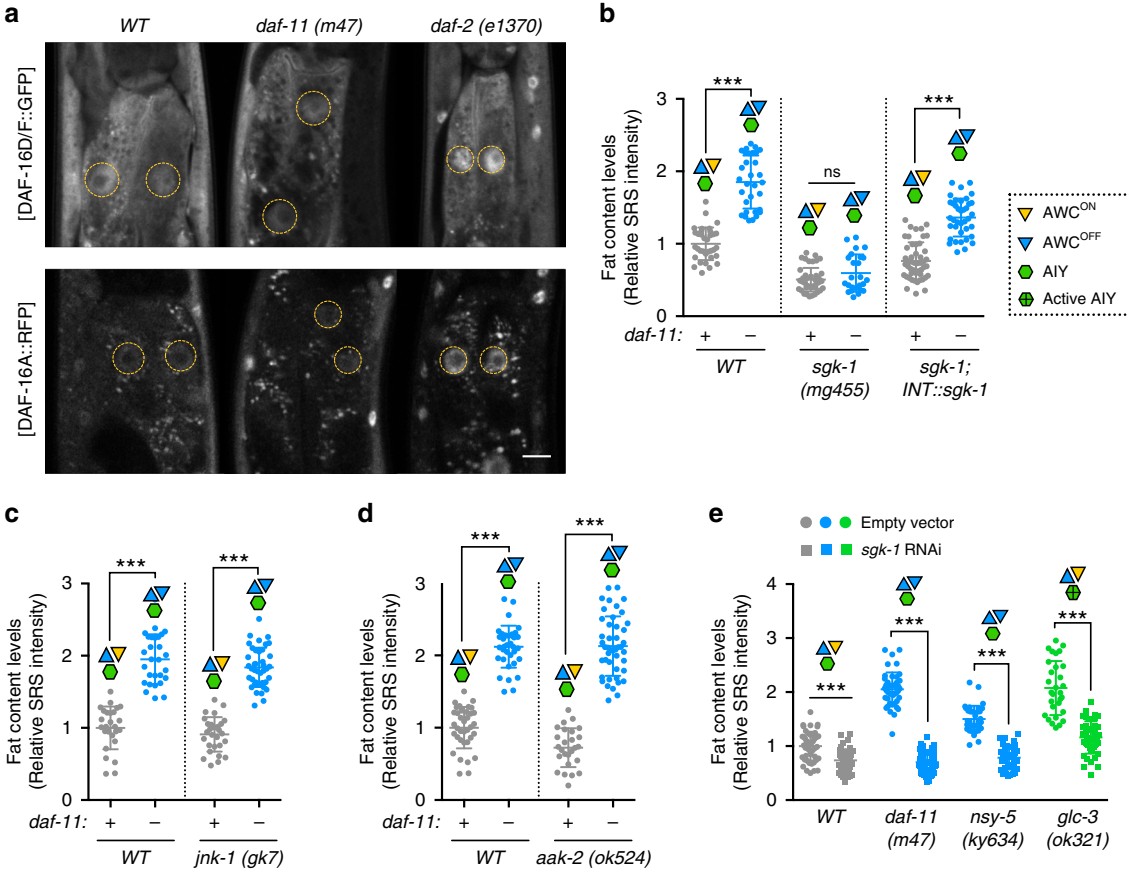

**Fig. 5 SGK-1 kinase regulates fat metabolism in response to olfactory signaling. a** Neither DAF-16A nor DAF-16D/F nuclear localization is strongly enhanced in the *daf-11* mutants, as it does in the *daf-2* mutants. Scale bar = 40 µm, yellow dashed lines circle nuclei of the intestinal cells. **b** Inactivation of SGK-1 fully suppresses the increased fat storage in the *daf-11* mutants. Intestine-specific restoration of *sgk-1* rescues the fat storage increase in the *daf-11*; *sgk-1* double mutants. Representative SRS images are in Supplementary Fig. 7c. **c, d** Neither the JNK homolog *jnk-1* (**c**), nor the AMPK homolog *aak-2* (**d**) affects the fat storage level in WT or in the *daf-11* mutants. Representative SRS images are in Supplementary Fig. 7d, e. **e** Knockdown of *sgk-1* in the *daf-11* and *nsy-5* mutants, that lack AWC$^{ON}$ neurons, or in the *glc-3* mutants with active AIY neurons suppresses the increased fat storage. Representative SRS images are in Supplementary Fig. 7a. For **b**–**e** data are mean ± s.d., ***$P < 0.001$, n.s. not significant by two-way ANOVA with Sidak's multiple comparison test. Numbers of animals used are listed in Supplementary Data 1. For **b**–**e** source data are provided as a Source Data file.

in the AWC defective *daf-11* mutants (Fig. 5a). We further discovered that inactivation of *sgk-1*, but not *jnk-1* or *aak-2*, fully suppresses the fat storage increase in the *daf-11* mutants (Fig. 5b–d and Supplementary Fig. 7c–e), and restoration of *sgk-1* expression specifically in the intestine rescues the fat storage increase (Fig. 5b and Supplementary Fig. 7c). Moreover, inactivation of *sgk-1* fully suppresses the fat storage increase in the *nsy-5* mutants with no AWC$^{ON}$ olfactory neurons and in the *glc-3* mutants with activated AIY interneurons (Fig. 5e and Supplementary Fig. 7a). Together, these results suggest that downstream of olfactory signaling, FOXO/DAF-16 is activated by SGK-1 to regulate lipid homeostasis cell-autonomously in the peripheral fat storage tissue.

**Neuropeptides link olfactory circuit and fat metabolism.** To investigate the endocrine factors that mediate the communication between the olfactory neural circuit and the fat storage tissue, we focused on neuropeptides that are small sequences of amino acids secreted from neurons and crucial for long-range signal transduction in animals. The carboxypeptidase E homolog, *egl-21* and the CAPS homolog, *unc-31* are required for neuropeptide processing and secretion, respectively[41,42]. We found that inactivation of either *egl-21* or *unc-31* fully suppresses the increased fat storage in the AWC defective mutants (Fig. 6a and Supplementary Fig. 6k).

Based on these results, we hypothesized that the neuropeptide release from AIY neurons mediates the olfactory regulation of peripheral fat storage. This hypothesis is supported by the opto-genetic stimulation of AIY neurons to directly induce neuropeptide signaling. As a photo-inducible adenylyl cyclase, the activation of bPAC by light induces dense core vesicle release and consequently neuropeptide signals[32]. We found that the photo-activation of bPAC in AIY neurons increases fat storage (Fig. 3h and Supplementary Fig. 6h).

We then searched for specific neuropeptides that selectively relay the information between AIY interneurons and fat storage cells. FLP-1 and FLP-18 are two neuropeptides expressed in AIY neurons[43]. We discovered that deletion of *flp-1*, but not *flp-18* suppresses the increased fat storage in the AWC defective mutants (Fig. 6b and Supplementary Fig. 6l), which can be rescued by the restoration of *flp-1* only in AIY neurons (Fig. 6c and Supplementary Fig. 6m). Inactivation of *flp-1* also suppresses the fat storage increase induced by 2-butanone odor exposure (Fig. 6d and Supplementary Fig. 6n). Conversely, overexpression of *flp-1* specifically in AIY neurons is sufficient to increase fat storage levels (Fig. 6c and Supplementary Fig. 6m). Together, these results suggest that the AIY-derived FLP-1 neuropeptide acts as a specific endocrine factor in fat metabolic regulation, which integrates the communication between the olfactory neural circuit and peripheral tissues (Fig. 6e).

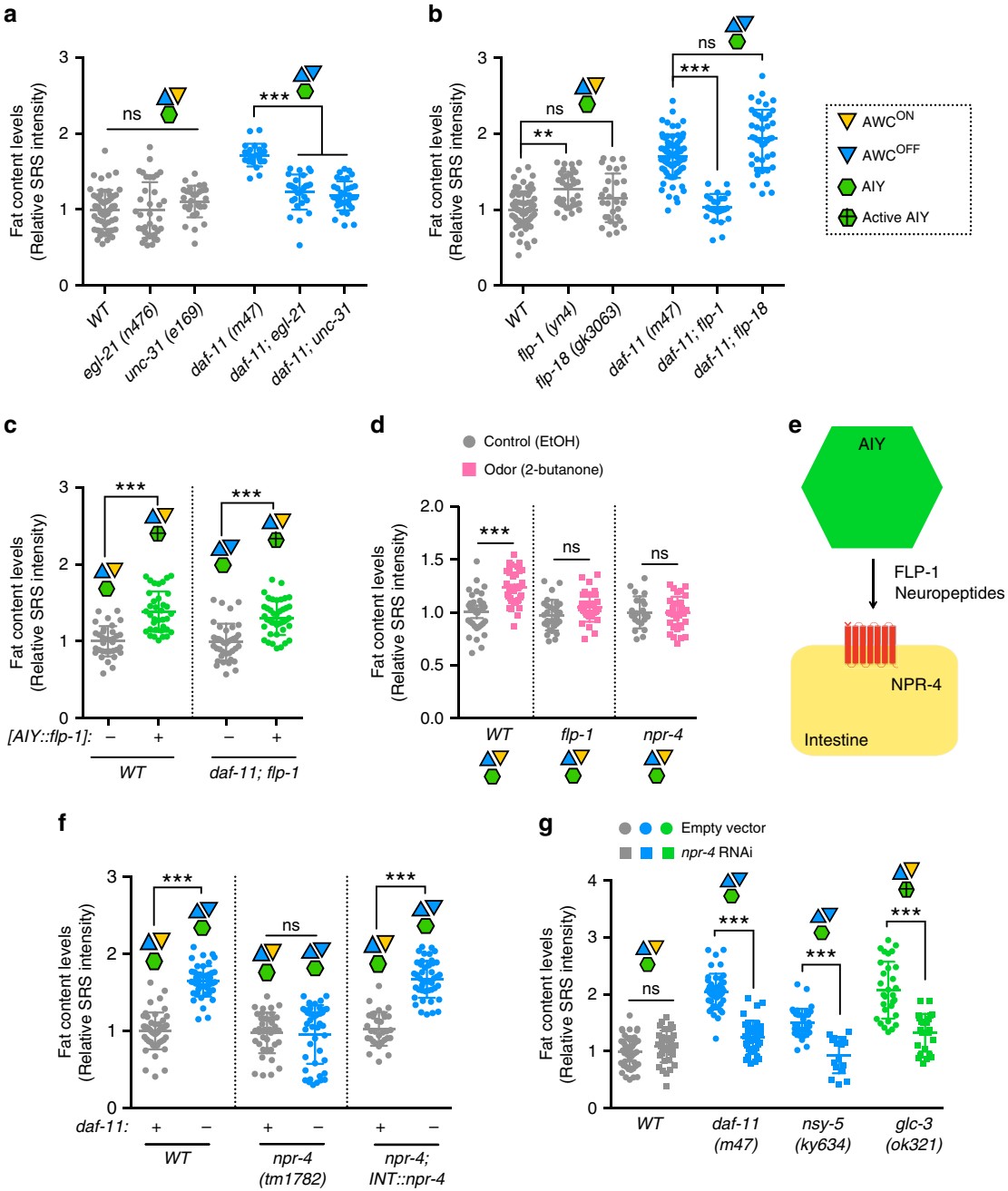

**Fig. 6 Neuropeptide signals link olfactory neural circuit and fat metabolism. a** Inactivation of either neuropeptide processing enzyme *egl-21*–carboxypeptidase E homolog, or neuropeptide secretion factor *unc-31*–CAPS homolog suppresses the increased fat storage in the *daf-11* mutants. Representative SRS images are in Supplementary Fig. 6k. **b** Among the neuropeptides that are secreted from AIY interneurons, deletion of *flp-1*, not *flp-18*, suppresses the increased fat storage in the *daf-11* mutants. Representative SRS images are in Supplementary Fig. 6l. **c** AIY-specific overexpression of *flp-1* increases fat storage under WT conditions. Restoration of *flp-1* expression only in AIY neurons is sufficient to rescue fat storage increase in the *daf-11; flp-1* double mutants. Representative SRS images are in Supplementary Fig. 6m. **d** Mutation of either *flp-1* neuropeptide or its putative receptor *npr-4* fully suppresses the fat storage increase conferred by the 2-butanone odor exposure. Representative SRS images are in Supplementary Fig. 6n. **e** Scheme of FLP-1 neuropeptides released from AIY interneurons acting through their putative receptor, NPR-4 that cell-autonomously regulates fat levels in the lipid storage tissue. **f** Inactivation of the *npr-4* neuropeptide receptor suppresses the fat storage increase in the *daf-11* mutants. Restoration of *npr-4* expression specifically in the intestine is sufficient to increase the fat storage in the *daf-11; npr-4* double mutants. Representative SRS images are in Supplementary Fig. 6o. **g** NPR-4 is required for the increased fat storage in the *daf-11* and the *nsy-5* mutants, that lack AWC^ON neurons, and in the *glc-3* mutants with active AIY neurons. Representative SRS images are in Supplementary Fig. 7a. For **a** and **b**, data are mean ± s.d., **$P < 0.01$, ***$P < 0.001$, n.s. not significant by one-way ANOVA with Tukey's multiple comparison test. For **c**, **d**, **f**, **g** data are mean ± s.d., ***$P < 0.001$, n.s. not significant by two-way ANOVA with Sidak's multiple comparison test. Numbers of animals used are listed in Supplementary Data 1. For **a**–**d**, **f**, **g** source data are provided as a Source Data file.

In searching for receptor(s) responding to the neuropeptide signal in fat metabolic regulation, we conducted an RNAi screen that targeted 71 different neuropeptide GPCRs[44] (Supplementary Table 1). We identified 5 candidate GPCRs whose inactivation suppresses the fat storage increase in the *daf-11* mutants: RNAi knockdown of *npr-4* or *ser-2* shows full suppression, whereas RNAi of *npr-6, tyra-3* or *frpr-10* shows partial suppression (Supplementary Fig. 1c). We also confirmed that genetic deletion of either *npr-4* or *ser-2* fully suppresses the fat storage increase in the *daf-11* mutants (Fig. 6f and Supplementary Figs. 5a, 6o). Among these candidates, we further focused on NPR-4, which is a homolog of mammalian neuropeptide-Y receptor and predominantly expressed in the fat storage tissue, the intestine[45]. We found that when *npr-4* expression is specifically restored in the intestine, the fat storage increase is rescued in the *daf-11; npr-4* double mutants (Fig. 6f and Supplementary Fig. 6o), which suggests the cell-autonomous effect of NPR-4 on fat metabolic regulation. Moreover, inactivation of *npr-4* suppresses the fat storage increase conferred by 2-butanone odor exposure (Fig. 6d and Supplementary Fig. 6n), AWC deficiency (Fig. 6g and Supplementary Fig. 7a), AIY activation (Fig. 6g and Supplementary Fig. 7a) or *flp-1* overexpression (Supplementary Fig. 5b). Together, these results suggest that NPR-4 acts downstream of AIY neuroendocrine signaling to regulate peripheral fat storage.

Furthermore, we found that in the peripheral tissue, NPR-4 acts upstream of FOXO/DAF-16 to regulate fat storage. In the AWC defective mutant with *npr-4* inactivation, the fat storage increase can be rescued by restoration of *npr-4* specifically in the intestine (Fig. 6f and Supplementary Fig. 6o), which, however, is suppressed by the *daf-16* inactivation (Supplementary Fig. 5c). Together, these data reveal the communication between an olfactory neural circuit and fat storage tissues via a specific neuroendocrine signaling pathway (Fig. 7).

## Discussion

Olfactory sensation plays crucial roles in regulating organism fitness and survival. AWA and AWC olfactory neurons are known to regulate organism lifespan[46], AWB olfactory neurons modulate reproductive longevity in response to microbial inputs[47], and more recently, AWC olfactory neurons are linked to the proteostasis control in peripheral tissues[48]. Here, our studies discover that olfactory perception through AWC neurons also actively regulates peripheral adiposity. We not only delineate the neuroendocrine mechanism linking the olfactory circuit to peripheral transcriptional control, but also uncover olfactory specificity as the key aspect of this regulatory modality. Two AWC neurons carry different olfactory receptors and respond to different environmental odors. We demonstrate that only AWC^ON neuron and its corresponding environmental odor can directly control the rate of lipid catabolism in peripheral fat storage tissues. Fat storage is a dynamic process, tightly balanced between lipid synthesis and catabolism. An alteration in total fat storage can be a result of changing either synthesis or catabolism, or both. In vivo tracking and distinguishing these two processes were difficult using canonical methods. Combining SRS microscopy with deuterium-labeled lipid pulsing and chasing, we are now able to directly assess the synthesis and catabolism rates of lipids in live animals[20]. We found that in the mutants with altered olfactory specificity, the rate of lipid catabolism is decreased, whereas the rate of lipid synthesis remains unchanged. As a result, these mutants accumulate increased amount of lipids in their peripheral fat storage tissues. Interestingly, these mutants do not show a decrease in their basal energy expenditure, implying a metabolic reprograming toward the consumption of non-lipid sources, such as carbohydrates and/or amino acids. This metabolic reprogramming

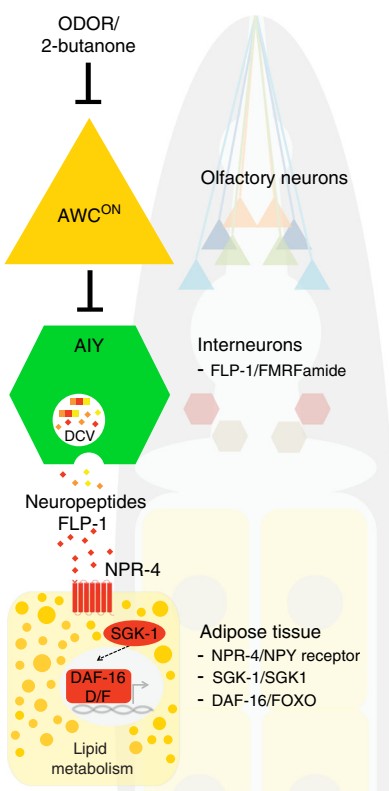

**Fig. 7 Olfaction regulates lipid metabolism through neuroendocrine signaling mechanisms.** Between two asymmetric AWC olfactory neurons, only AWC^ON regulates peripheral lipid metabolism. In response to specific environmental odors, this olfactory neuron signals through a specific downstream neural circuit and neuroendocrine signaling pathway (conserved components in mammals are indicated) to directly control lipid homeostasis in peripheral metabolic tissues. DCV, dense core vesicles.

triggered by olfactory inputs is likely relevant to energy homeostasis in a broad range of organisms. Upon receiving different olfactory inputs from the environment, organisms may apply different metabolic strategies and switch the preference for different energy sources to support their survival and reproduction.

Interestingly, among several odorants that we have tested so far, only 2-butanone exposure sufficiently modulates fat storage dynamics. In the environment, *C. elegans* are exposed to hundreds of odors that bacteria emit, including 2-butanone, a methyl ketone[49]. It is known that under certain biological conditions, some fungi and bacteria can produce 2-butanone by decarboxylation of β-keto acids that are formed during the process of fatty-acid oxidation[50]. Thus, the production of 2-butanone may indicate reduced availability of environmental lipids. On the other hand, 1-octanol is a product of glucose breakdown pathways[51] and benzaldehyde can be produced from amino acid, phenylalanine[52], which are not associated with fatty-acid catabolism and have no effects on fat storage dynamics. Therefore, 2-butanone may serve as a specific volatile cue signaling environmental lipid scarcity to animals, and trigger the adjustment of animals' metabolism toward fat accumulation. This way, animals could be better prepared for the shortage of lipid resources that are essential for successful reproduction. Additionally, 2-butanone affects bacterial motility and biofilm formation[53], and thus may also serve as a signal associated with bacterial stress to modulate fat storage dynamics in the host. Recent studies have shown that AWC neurons regulate peripheral proteostasis in response to bacterial odors, and discovered specific NLP neuropeptides involved in this regulation[48]. For this olfactory regulation, it remains unclear the

neural circuit and peripheral factors that transduce signals from AWC neurons to regulate proteostasis. It would be interesting to investigate whether the neuroendocrine signaling pathway that we delineated here can also regulate proteostasis in future studies, given the close interaction between endoplasmic reticulum stress and obesity[54]. On the other hand, different neuropeptides are implicated in the olfactory regulation of peripheral proteostasis (NLP) and adiposity (FLP). Thus, it is also possible that AWC olfactory neurons act through different neural circuits and neuroendocrine mechanisms to regulate different physiological activities in the periphery, and these regulations cooperate to ensure organism fitness in response to environmental fluctuation. Together, these studies support the significance of olfactory modality for animals to anticipate environmental changes and adjust their metabolic activities accordingly. These findings also provide evidences for the crucial role of cephalic phase in regulating metabolism. Sensory perception of food by visual, auditory, and olfactory systems induces heart rate and increases the secretion of digestive enzymes even before ingestion[55]. These responses known as cephalic phase enable efficient nutrient intake and metabolic adaptation, through modulating specific hypothalamic neurons and consequently priming physiological responses in peripheral organs[55,56]. These regulations are crucial to maintain organism health under physiological conditions.

Pathologically, obesity is commonly associated with weak olfactory identification[57] and patients with anorexia nervosa have enhanced smell capacity[58]. Clinical studies do not allow us to distinguish whether these changes in olfactory acuity are the cause or the consequence of metabolic dysfunction. Animal studies demonstrated that mice fed with high-fat diet show diminished smell capacity as a result of olfactory sensory neuron loss[59]. It is also shown that the voltage-gated K$^+$ channel (Kv) subtype Kv1.3 knockout mice have increased olfactory sensitivity and are resistant to diet-induced obesity[60], but surgical removal of the olfactory bulb prevents this resistance to obesity upon high-fat diet[4]. In contrast, recent studies showed that the ablation of olfactory sensory neurons protects mice from weight gain when fed a high-fat diet, and increasing olfactory sensitivity by IGF-1 knockout in the olfactory epithelium leads to lipid accumulation[3]. The discrepancy of these findings might be because crude manipulation of the olfactory system using different strategies alters olfactory specificity and sensitivity in a different manner. As demonstrated in our studies, olfactory specificity is crucial for lipid metabolic regulation, which can be determined at the levels of environmental odors, olfactory neurons and neuroendocrine signaling pathways.

The olfactory system carries high complexity and specificity. Odors are complex environmental stimuli, and discrimination of different odors depends on the combinatorial action of a large number of diverse odor receptors. For each individual, differential combination of odor receptors in olfactory neurons provides a characteristic fingerprint of smell identity and capacity. Our studies reveal that not all environmental odors affect fat metabolism, and for the same type of olfactory neurons, even presenting different receptors (AWC$^{ON}$ vs. AWC$^{OFF}$) can lead to distinct effects on fat metabolism. Therefore, the characteristic olfactory specificity and diversity among individuals may be associated with their unique lipid metabolic responses to environmental stimuli and consequently their distinctive susceptibility to obesity. Interestingly, olfactory receptor polymorphisms have been associated with the onset and severity of obesity through genome-wide association studies[6]. Therefore, further mammalian studies should focus on the mechanistic link between olfactory specificity and peripheral adiposity, which would provide ways in preventing and treating obese patients.

Biogenic amines, including octopamine, tyramine, dopamine, and serotonin are crucial neurotransmitters and/or neuromodulators, which modulate various physiological and behavioral responses in *C. elegans*. In particular, serotonin signaling is known to adjust food intake and maintain energy homeostasis based on nutrient availability[9,61,62]; octopamine is shown to be involved in the neuronal regulation of longevity and mitochondrial dynamics[63]; and dopamine has been reported to affect fat mobilization[64]. However, the role of tyramine in metabolic regulation remains unclear. Interestingly, in our genetic screen searching for neuropeptide receptors responding to neuroendocrine signals from the AWC-AIY circuit, we also discovered SER-2, a putative tyramine receptor, in addition to NPR-4. We found that SER-2 is required for the increased fat storage in the *daf-11* mutants (Supplementary Figs. 1c and 5a). SER-2 is expressed in a certain subset of neurons and pharyngeal cells and regulates foraging behavior in *C. elegans*[65,66]. Thus, we do not think that SER-2 acts in the intestine as a peripheral receptor, but we expect its involvement in the modulation of the AWC-AIY olfactory neural circuit. Understanding this modulatory role of SER-2 and tyramine signaling in controlling peripheral fat storage will be an interesting topic for future characterization.

Our studies also reveal the post-developmental role of cGMP signaling in regulating fat metabolism. During development, *daf-11* plays crucial roles in controlling dauer formation by acting mainly in ASJ gustatory neurons[22]. Unexpectedly, we discovered that the fat storage regulation by *daf-11* at adulthood is exclusively mediated by its function in AWC olfactory neurons and acts through a unique neuroendocrine signaling pathway, which is completely distinct from its dauer regulatory mechanisms. In response to environmental insults, alterations in developmental strategies can ensure successful survival. However, mediators of these adaptations during development can also contribute to the onset of adult diseases. Our studies reveal how a regulator of developmental plasticity influences metabolic balance in adulthood, which may shed light into understanding the developmental origin of common adult-onset obesity.

## Methods

**C. elegans strains and transgenics**. The nematodes were mostly obtained from the *Caenorhabditis* Genome Center (CGC). Some nematode strains were gifts from other labs or generated in the lab. Full list of *C. elegans* strains used in this study is given in (Supplementary Data 2). The lab strains that were obtained from CGC or other labs were backcrossed at least three times to the wild-type control, N2 in our laboratory. Worms were maintained on standard nematode growth medium (NGM) agar plates seeded with pre-cultured bacterial strains unless otherwise noted. Chemosensory mutants used for screening altered lipid levels were grown at 20 °C and imaged at L4 stage. For most of the experiments unless indicated, worms were maintained and grown at permissive temperature 15 °C until L4 stage and shifted to restrictive temperature 25 °C and grown to 3-day-old adults for imaging. For the biochemical assay to measure TAG levels and qPCR analysis to measure *sod-3* expression levels, L4 worms were used to avoid the possible interference from eggs. For *egl-21*, *unc-31* genetic interaction experiments, L4 worms were used to avoid internal hatching of eggs in adults. For deuterium-labeling experiments, the whole assay last 3 days. So the rate of lipid mobilization was in fact analyzed when worms are at day 3. For DAF-16::GFP, its nuclear localization is enhanced in wild-type worms at 25 °C and at day 3, even without *daf-2* or *daf-11* inactivation. So we have shifted the worms to 25 °C at L4 and moved them back to 20 °C for analysis at day 1.

**Molecular cloning and generating transgenics**. Promoter lengths and their primer sequences used to generate transgenics are given in (Supplementary Data 3). Most of the transgenic arrays were generated using the Multisite Gateway System (Invitrogen). Promoters were cloned to 1′position Donor Vector pDONR 221/P1-P4; cDNAs were cloned to 2′position pDONR 221/P4r-P3r and sl2::RFP or GFP::unc-54 3′UTR was cloned to 3′position pDONR 221/P3-P2 using Gateway BP reaction. For expressing DAF-16D/F::GFP fusion in the intestine (MCW441) we cloned *GFP::unc-54 3′UTR* without SL2 sequence to the 3′position. Entry vectors were then recombined into destination vector, pCMP1 (pCFJ151 modified to contain Gateway Pro LR recombination sites) using Gateway LR reaction.

For generating transgenic line MCW711, we PCR amplified the whole genomic region of *daf-11*, 12 kb fragment including the upstream promoter and coding sequence, and used this fragment for performing fusion PCR together with *sl2::RFP::unc-54 3′UTR*. For generating the transgenic line MCW1037, we performed fusion PCR to combine *str-2* promoter with *ChR2::sl2::RFP::unc-54 3′UTR*.

Transgenic strains were generated by injecting the DNA mixture into the gonads of young adult worms using the standard *C. elegans* microinjection protocol. For all of the transgenic lines, we injected 10 ng/μl transgenic construct, 10 ng/μl co-injection marker and 80 ng/μl salmon sperm DNA. For the generation of integrated strains, late L4 stage strains containing extrachromosomal arrays were exposed to 4000 rads of gamma irradiation in 5.9 min, and integrated strains were backcrossed to wild-type N2 at least 5 times.

**SRS microscopy, sample preparation, and image quantification**. A detailed protocol on how to setup SRS microscopy, to prepare worms for imaging and to quantify images was described in ref. [16]. For SRS microscopy, spatially, and temporally overlapped pulsed Pump (tunable from 720 to 990 nm, 7 ps, 80 MHz repetition rate) and Stokes (1064 nm, 5~6 ps, 80 MHz repetition rate, modulated at 8 MHz) beams provided by picoEMERALD (Applied Physics & Electronics) were coupled into an inverted laser-scanning microscope (IX81, Olympus) optimized for near-IR throughput. A ×20 air objective (UPlanSAPO, 0.75 N.A., Olympus) was used for imaging fat storage. A ×60 water objective (UPlanAPO/IR, 1.2 N.A., Olympus) was used for $CD_2$ signal imaging. After passing through the sample, the forward going Pump and Stokes beams were collected in transmission by an air condenser. A high OD bandpass filter (890/220, Chroma) was used to block the Stokes beam completely and to transmit only the Pump beam onto a large area Si photodiode for the detection of the stimulated Raman loss signal. The output current from the photodiode was terminated, filtered, and demodulated by a lock-in amplifier (HF2LI, Zurich Instruments) at 8 MHz to ensure shot noise-limited detection sensitivity. $CH_2$ signals were imaged at 2845 cm$^{-1}$, $CD_2$ signals were imaged at 2110 cm$^{-1}$. The microscope was controlled by Olympus Fluoview 1000 software. SRS microscopy images were quantified using ImageJ software (NIH). Polygon selection tool was used to select the area to be quantified and average pixel intensity was calculated with the "analyze-measure" command. After subtracting the background intensity, all measurements were averaged to obtain mean and standard deviation. In each imaging session, ~20–30 worms were immobilized with 1% sodium azide on 2% agarose pads on glass microscope slides. Mutant lipid levels were normalized to those of wild-type worms. Numbers of worms used for SRS microscopic imaging are given in Supplementary Data 1. Representative SRS images are shown in (Supplementary Figs. 6 and 7). In all the SRS microscopy results shown in this manuscript, the imaging parameters were setup for neutral lipids stored in lipid droplets. Under this condition, membrane lipids or free fatty acids do not reach a high enough local concentration to be detected by SRS.

**Biochemical assay**. At least 5000 age-synchronized worms were grown at the permissive temperature, 15 °C, till L3 stage for 40 h, then switched to the restrictive temperature, 25 °C for 8 h, and harvested at L4 stage. At 15 °C, less than 10% of the *daf-11* mutants become dauers and we hand-picked and removed them before collecting L4 worms for further biochemical assay. After collection, the worms were ground with a pestle and sonicated to lyse worm cells. Using the worm lysate, total lipid levels were measured using Triglyceride Colorimetric Assay Kit (Cayman Chemical). Lipid levels were normalized to total protein levels, which were measured using Bio-Rad Protein Assay (Bio-Rad). Data shown represent five independent replicates.

**Deuterated fatty-acid supplementation**. OP50 bacterial culture was mixed well with 1 mM deuterated oleic acid (OA-D34, Sigma), and then seeded onto NGM plates and grown overnight. To analyze the rate of lipid synthesis, 1-day-old adult worms were fed with OA–D$_{34}$ supplemented bacteria and imaged using SRS microscopy at indicated time intervals. To analyze the rate of lipid catabolism, OA–D$_{34}$ labeled worms were transferred onto fresh NGM plates with unlabeled OP50 and imaged using SRS microscopy at indicated time intervals. SRS microscopy can separately image non-labeled fat using C–H bonds and labeled fat using C–D bonds. During the whole pulse-chase experiments, C–H signals have not changed, because (1) the animals are kept in a well-fed condition and the level of non-labeled fat is in equilibrium, and (2) deuterium-labeled fat is supplied at a low amount and does not take over non-labeled fat. In addition, C–D bonds cannot be converted into C–H bonds, or vice versa. In all the SRS microscopy results shown in this manuscript, the imaging parameters were setup for neutral lipids stored in lipid droplets. Under this condition, membrane lipids or free fatty acids do not reach a high enough local concentration to be detected by SRS. In particular, the lipid flux analyses only track neutral lipids synthesized from OA-D$_{34}$ in lipid droplets. Data shown in (Fig. 1f) represents one replicate with at least 5 worms for each genotype at every time point. Another replicate is shown in supplementary materials in (Supplementary Fig. 2c).

**Physical activity measurements**. On the first three days of adulthood, age-synchronized worms were recorded individually for their spontaneous movement for 1 min after tapping their plate using an SMZ1500 stereo microscope (Nikon) connected to a C11440 camera (Hamamatsu). Individual worms were tracked using NIS Elements AR imaging software (Nikon) and average velocity (μm/sec) was calculated. For pharyngeal pumping measurement, same setup was used. The movement of the terminal bulb of the pharynx in the posterior direction was counted as one pharyngeal pump. The number of pumps in each minute interval was calculated. To measure the duration of defecation cycle, the time between the contractions of posterior body muscles was measured simply by observation under the stereo microscope. For each worm, the duration of at least three defecation cycles was measured. For all physical activity measurements, at least 10 animals were used for each genotype and the measurements were repeated three independent times. For brood size analysis, 10 synchronized L4 hermaphrodite larvae were transferred to individual plates at 25 °C every 12 h and the number of progeny was counted until reproductive cessation. The total number of progeny per hermaphrodite was calculated. Mutants and wild-type worms were compared using one-way ANOVA with Tukey's multiple comparison test.

**High-resolution locomotion analysis**. Detailed analysis of *C. elegans* locomotion was performed using 1-, 2-, and 3-day-old adult animals exploring 12-well agar plates seeded fresh with OP50. Individual animals were recorded for 40 min by a 1024 × 1024 pixels Hamamatsu camera. For each worm, 6000 images were taken every 400 msec. The camera was controlled and the data were analyzed via the software iBeN (Imaging Behavior of Nematode) was described before[67]. The speed and the curvature were calculated from the trajectory of each worm during the whole video by iBeN using similar methods as described in ref. [68]. "Resting" refers to speed less than 5 μm/sec; "dwelling" represents speed higher than 5 μm/sec but less than 60 μm/sec and curvature more than 40° and "roaming" represents speed is higher than 60 μm/sec and curvature less than 40°. For these assays, at least three animals were used for each genotype and for each day of adulthood. Mutants and wild-type worms were compared using one-way ANOVA with Tukey's multiple comparison test.

**Oxygen consumption assays**. Oxygen consumption rate in wild-type worms and *daf-11* mutants were measured using Seahorse XF96 extracellular flux analyzer (Agilent Technologies), following the protocol described in ref. [69]. Measurements were performed at room temperature (27 °C with heat generated by the machine). Age-synchronized worms were grown on *E. coli* OP50 NGM plates and collected as 1-day-old adults. Worms were washed in M9 buffer twice to remove bacteria, and approximately 25 worms were added in 200 μl M9 buffer to the wells of XF96 microplate. The oxygen consumption rate was measured five times at baseline and five times after the addition of etomoxir sodium salt (Sigma-Aldrich), an inhibitor of mitochondrial β-oxidation via carnitine palmitoyltransferase 1, dissolved in M9 to a 500 μM final concentration. For each well, the percent change in the average oxygen consumption rate before and after etomoxir addition was used to calculate the oxygen consumption rate dependent on mitochondrial fatty-acid β-oxidation. At least 10 wells were used for each wild-type and mutants in each assay and the assays were repeated two independent times. Worms in each well were counted after the assay is done and the oxygen consumption rate is normalized to worm number. Statistical analysis was performed using the Student's t-test.

**DiI and DiO staining**. Approximately 200 L4-stage worms were stained with either with red DiI (1,1′-Dioctadecyl-3,3,3′,3′-Tetramethylindocarbocyanine Perchlorate, Life Technologies) or green DiO (3,3′-Dioctadecyloxacarbocyanine Perchlorate, Life Technologies) in 1.5 mL Eppendorf tubes on a rotator for 3 h. The working concentration was 15 μg/ml. The excess dye was removed by letting the worms crawl for 1–2 h on standard plates that are kept in a dark box. For imaging, 20–30 worms were immobilized with 1% sodium azide on 2% agarose pads on glass microscope slides.

**Confocal fluorescence microscopy imaging**. Same inverted laser-scanning microscope (IX81, Olympus) used for SRS microscopy imaging was also utilized for confocal imaging. The microscope has three continuous visible lasers (405, 488, and 559 nm) controlled by an Acoustic Optical Tunable Filter (AOTF). The 488 nm laser is used for *raxIs45[Pdaf-11::daf-11::sl2::GFP]* and DiO staining and 559 nm was used to image DiI staining in *C. elegans* chemosensory neurons in sequential scanning mode. A ×60 oil objective (PlanAPO N, 1.42 N.A., Olympus) was used. The images were taken using an AxioCam ICc3 camera (Zeiss) and the microscope was controlled by Olympus Fluoview 1000 software.

**RNAi feeding**. For RNAi-based experiments, RNAi clones from the libraries generated in the laboratories of Dr. Julie Ahringer and Dr. Marc Vidal were used. All the RNAi clones (*daf-16*, *sgk-1*, *npr-4* and other candidate neuropeptide receptor clones) used in this study were verified by sequencing. Furthermore, the corresponding mutants for most of the genes were used in each case to confirm the RNAi experiment results. A cherry pick RNAi library targeting neuropeptide

GPCRs was constructed based on the list that was published in ref. [44]. Worms were synchronized by adult-egg laying on plates with HT115 bacteria transformed with either empty L4440 vector or vector carrying RNAi against each neuropeptide receptor, and eggs were grown on these plates at 15 °C until larval L4 stage. Then, the plates were shifted to 25 °C, and 3-day-old adults were imaged using SRS microscopy. During the screen, at least four worms were imaged for each RNAi clone and the screen candidates were validated with three independent replicates. The combined results from these replicates are represented in (Supplementary Fig. 1c). The candidate receptor clones were verified by sequencing.

**Quantitative RT-PCR**. Total RNA was isolated from at least 1000 age-synchronized larval L4 stage worms using Trizol extraction with column purification (Qiagen). Synthesis of cDNA was performed using the amfiRivert Platinum cDNA Synthesis Master Mix (GenDEPOT). Quantitative PCR was performed using Kapa SYBR fast qPCR kit (Kapa Biosystems) in a 96-well Eppendorf Realplex 4 PCR machine (Eppendorf). All data shown represent three biologically independent samples and were normalized to rpl-32 as an internal control.

**Odor exposure**. Approximately 60 1-day-old adult worms were picked onto an empty NGM plate with minimal bacterial transfer. A glass coverslip was placed inside the lid of the plates. 2 µl of the odor, 2-butanone, 2,3-pentanedione, benzaldehyde, isoamyl alcohol, 1-octanol, or 2-nonanone (Sigma-Aldrich), was dropped onto the coverslip and plates were sealed tightly with Parafilm. Odor exposure plates did not have bacteria to promote lipid catabolism and to eliminate any interference from the food. All test and control plates were put inside a glass desiccation chamber (Pyrex) at 20 °C. After 4 h of odor exposure, half of the worms were imaged for lipid storage using SRS microscopy. Other half were transferred to standard NGM plates seeded with OP50 E. coli bacteria and imaged 4 h later for analyzing the restoration of lipid levels. At each time point and for each genotype, the fat storage levels were normalized to control plates, which did not have the odor.

**Optogenetics**. Ex[AWC$^{ON}$::ChR2] and Ex[AWC$^{OFF}$::ChR2] transgenics and N2/WT control worms were raised from egg hatching in the dark on NGM plates containing 500 µM all-trans-retinal[22]. Worms were synchronized by picking them at L4-stage in the dark room and 1-day-old adult worms were used for analysis. On day 1, worms were transferred to unseeded plates and exposed to 470 nm blue light from LEDs (intensity 10 µW/mm$^2$) for 1 h, then kept in dark for 1 h. Controls were kept in the dark for 2 h. After 2 h, worms were imaged by SRS microscopy for fat level analysis as previously described.

Ex[AIY::bPAC] transgenics and N2/WT control worms were raised on standard NGM plates seeded with OP50. They were age-synchronized at L4 stage and next day, 1-day-old adult worms were analyzed for fat level changes. On day 1, worms were transferred to unseeded plates and exposed to 470 nm blue light from LED with intensity 70 µW/mm$^2$ for 15 min per hour for 4 h. After blue light exposure, worms were imaged by SRS microscopy for fat level analysis as previously described.

**Quantification and statistical analysis**. No statistical method was used to predetermine sample size. Animals were randomly picked from experimental groups for analysis, and data from all the samples were used for statistics without exclusion. Investigators were not blinded. Student's t-test was used when comparing two samples. One-way ANOVA was performed with Dunnett's test when multiple samples were compared to a single control, and with Tukey's test when multiple samples were compared to each other. Two-way ANOVA was performed with Sidak's ad-hoc test when two effectors, genotype and RNAi, were compared. Fisher's exact test was used to compare the dauer formation across different mutant backgrounds. All statistical analysis was done in PRISM 7.

**Reporting summary**. Further information on research design is available in the Nature Research Reporting Summary linked to this article.

## Data availability

All relevant data generated or analyzed during this study are included in this manuscript and/or its supplementary information, or can be obtained from the corresponding authors upon request. The data underlying Table 1, Figs. 1a–g, 2b–d, 3a, c–h, 4a–d, 5b–e and 6a–d, f, g and Supplementary Figs. 1a, b, 2a–f, 3a–c, 4a–c and 5a–c are provided as Source Data.

## Code availability

The codes used for high-resolution behavioral analyses are available at https://github.com/mengwanglab/iben.

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

## Acknowledgements

We thank Y. Yu and D. Deng for experimental support, and B. Arenkiel and H. Dierick for critical reading of the manuscript. We thank W. Steuer Costa and A. Gottshalk for sharing the information and plasmids of bPAC. This work was supported by NIH grants R01AG045183 (M.C.W.), R01AT009050 (M.C.W.), R01AG062257 (M.C.W.), DP1DK113644 (M.C.W.), March of Dimes Foundation (M.C.W.), Welch Foundation (M.C.W.), and by HHMI investigator (M.C.W.), HHMI international pre-doctoral student fellow (A.S.M.).

## Author contributions

A.S.M. and M.C.W. designed the study; A.S.M., S.M.G., and H.Z. performed the experiments; and A.S.M. and M.C.W. wrote the manuscript.

## Competing interests

The authors declare no competing interests.
