## [Peer Review File · Nature Communications]

Reviewers' comments:

Reviewer #1 (Remarks to the Author):

Mutlu et al. describe that olfaction can regulate fat storage and describe a circuit from an asymmetric olfactory neuron, via an interneuron, a neuropeptide and its receptor, eventually funneling into daf-16/FoxO activation. I think the manuscript is a good fit for Nature Communication, provided that one point gets clarified by the authors. I'm not quite understanding how the authors explain that during this olfactory information processing daf-16 does NOT appear to translocate to the nucleus in the intestine. If it doesn't translocate to the nucleus, how can it fulfill its "canonical" role on controlling fat metabolism in the intestine? I'm perhaps overlooking something here and hope that the authors can clarify this matter.

Reviewer #2 (Remarks to the Author):

This manuscript by Mutlu et al. describes a novel genetic pathway by which the olfactory neuron AWC in *C. elegans* modulates peripheral fat metabolism through downstream endocrine signaling and not via changes in food intake. The authors start with a broad array of chemosensory mutants expected to act in the nervous system, identifying alterations in fat metabolism by the use of state-of-the-art Raman scattering microscopy in daf-11 mutants. The authors go on to establish a role of the AWC neurons in reciprocal regulation of body fat mass, and an elaborate signaling pathway from AWC to the AIY interneuron finally to the intestine via neuropeptide flp-1 signaling, possibly engaging the NPY-receptor ortholog NPR-4. The experimentation is compelling and rigorously conducted, and the manuscript is well written and clear, and will be of broad interest to the readership of Nature Communications after some important points are addressed.

Major comments:

The manuscript alludes to but does not do justice to the myriad literature on the so called "cephalic" phase of food sensing, which is known to modulate peripheral metabolism. While this likely involves both visual perception and olfaction, a very nice paper detailed such a mechanism in *Cell* in 2018 (PMC6541012). Studies such as this one should be referred to in the manuscript to bring the non-expert reader up to speed on the state of the art in the literature. Interestingly in *C. elegans* which does not have an obvious visual system, the cephalic phase is likely to be dominated by the olfactory system.

The analyses of fat mass could be confounded by the presence of a moderate level of constitutive dauer formation (known of course to drive high fat mass and reduced fat oxidation). Do the presence of dauers alter interpretation of the data in Figure 1 (particularly the "grind up" analyses as done in Fig 1b? To this end the data for Figure 2D ASJ rescue are reassuring. The data shown are for day 3 adults. Are the increases in fat manifest at earlier developmental stages? Do the daf-11 mutants have a fecundity defect? Fecundity is known to impact lipid stores in the worm.

The lipid flux analyses are excellent—but can the authors be more specific about which "lipids" they are tracking by SRS microscopy in the deuterium labeling experiments? Are we talking about all lipids into which oleate is incorporated (membrane and neutral lipids) or mainly just neutral lipids? I would guess the latter given their predominance but it would be good to have an estimate of the preponderance of triacylglycerols in the assay.

Data in figure 2C and D are improperly statistically analyzed. The comparison should be by 2-way ANOVA with post-hoc testing as the experiment has two variables—strain, and presence or absence of the rescue transgenic array. The same criticism holds true for all experiments with two variables (Fig 3D, E, F, H; 4c; 5B, C, D, E; 6C, D, F, G) which should be analyzed by two-way ANOVA

The data on SGK-1 based activation of DAF-16 is in contrast to what is thought to be the canonical role of AGC family kinases in inhibition of FOXO activity, particularly with AKT and SGK. However, these data are in line with work on lifespan extension in *sgk-1* gain of function mutants and how this depends upon *daf-16*. The authors may want to draw attention to this work (PMC3824081).

The authors are somewhat dismissive of the *ser-2* result that fully reverses the high fat mass in the *daf-11* mutant. *Ser-2* is a putative biogenic amine receptor, others have worked on signaling of serotonin to the intestine and regulation of body fat mass. While *ser-2* is a putative tyramine receptor, the authors may want to put this work in context.

In general, the direct connections of FLP-1 to NPR-4 are lacking, and the mechanistic connections, while possible, are missing. Further, the idea that multiple connections (based not in the least on the authors' *ser-2* result) from AIY to intestine may govern fat mass raise the possibility that NPR may be a separate sufficiency signal not necessarily engaged directly by FLP-1. Without evidence of a FLP-1 overexpression phenotype that is mitigated by loss of NPR-4 signaling (and even then the connection could be indirect), the conclusions and implications drawn from the model (Figure 7) about direct connections should be softened somewhat.

Minor comments:

- 1) *C. elegans* do not have adipose tissue, and it is confusing to call intestine adipose tissue (line 109). Please rephrase to indicate that intestine is at least one of several major fat storage sites in the worm.
- 2) Can the authors speculate as to why *daf-11* mutants *ks67* and *m47* show different intensity of fat mass phenotypes?
- 3) It is probably an overstatement to say that food intake is not different based upon only pharyngeal pumping rates (lines 113-122). The authors may want to indicate the limitations of such an assay in truly assessing food intake.

Reviewer #3 (Remarks to the Author):

This manuscript uses a sophisticated set of genetic tools to investigate how specific odors and olfactory neurons in *C. elegans* regulate fat metabolism. Taking advantage of a substantial body of previously established knowledge about olfaction and fat metabolism in the worm, and using an impressive set of genetic tools that the authors and others have established previously, the manuscript is able to delineate a detailed mechanism by which an odorant acts through a series of cells and signaling molecules to ultimately affect fat storage in the intestine. Overall, the work is well done, and the manuscript describes an impressive amount of work and results. The work is significant and of broad interest in that it provides a novel level of mechanistic insight into how odorants affect fat metabolism. The results are convincing and well supported by the experimental evidence presented. The statistical analysis of the data shown is appropriate.

I list below three "major criticisms", but the authors should be able to deal with these easily by revising their writing. Three additional minor points should also be easily dealt with.

Major criticisms.

1. The results on the locomotion behavior of *daf-11* mutants are confusing and I was not satisfied with how they were handled. I go into this issue in detail here because the authors conclude from their locomotion analysis that *daf-11* likely does not affect fat storage by decreasing locomotion. I was not convinced by their argument. The relevant results are presented in a paragraph in the Results in lines 113-122, in Fig 1c-e, and in Supplemental Fig 2a, 2b. The confusion results from the fact that two measurements of locomotion (Fig 1c and Supplemental Fig 2a) show no change in locomotion in *daf-11*, while another measurement (Supplemental Fig 2b) shows that *daf-11* mutants spend only about 1/3 as much time as the wild type in the high-locomotion "roaming"

state. The definition of roaming includes a wide range of locomotion rates, so it is conceivable that there is no contradiction between the results shown in the three analyses. It is also possible that there really is a contradiction: maybe the data set collected for Fig 1c (a small number of worms analyzed for 1 minute each) was insufficient to properly sample worms in both the roaming and dwelling states. At the very least, the authors should provide a plausible explanation of why one measure of locomotion shows a large change in *daf-11* while two other measurements do not. In interpreting their results, the authors cite a paper for the result that *che-2* mutants also had decrease in roaming, and in the supplemental Fig 1a of this manuscript, a *che-2* mutant showed no change in fat content. The reader does not know if the same mutant allele of *che-2* was used in both analyses and whether the magnitude of the roaming changes in *che-2* are comparable to those in *daf-11*. So, I found this argument unsatisfying.

2. In Figure 4 and Figure 6, the schematics at the top of the panels depict schematically that the *daf-11(m47)* mutation results in worms having two AWCON neurons (graphically depicted as two blue triangles). This confused me, as I do not believe such an effect of *daf-11* is cited as having been established in any reference or established by any results presented earlier in the manuscript. The earlier results in this manuscript do show that *daf-11* results in fat storage defects similar to those of the *nsy-5* mutant that is established to result in the presence of two AWCON neurons, but the AWCON identity is defined by the expression or non-expression of specific molecular markers (*str-2* and *srsx-3*) that were not examined in the *daf-11* mutant. The use of the blue and orange triangles to depict AWC type should be corrected or clarified.

3. The experiments in Figure 6 establish that NPR-4 acts in the intestine to mediate downstream effects of *daf-11* and AWCON on fat storage. I agree that these data strongly infer that that NPR-4 may be a receptor for the neuropeptide FLP-1. However, the current practice among people studying *C. elegans* neuropeptides is to not conclude definitively that a GPCR is a receptor for a neuropeptide based only on such indirect genetic evidence. The additional experiment required to make such a conclusion would be to express the GPCR on heterologous cells and show that application of the peptide activates the receptor. The authors did not do that experiment. Therefore, the wording used to talk about the hypothesis that NPR-4 may be the FLP-1 receptor needs to be chosen carefully. In the Results (lines 365-366) the authors are appropriately cautious in their wording. I would appreciate it if in the Figure 6d and 6e legend, the authors also are more cautious, i.e. instead of stating that NPR-4 is the FLP-1 receptor, the statement should be that NPR-4 is the putative or apparent FLP-1 receptor.

Minor points

1. The authors should search the manuscript and supplemental material for all occurrences of "*C. elegans*" and "*Caenorhabditis*" and make sure they are all italicized.

2. In supplemental Fig 2, panels B and C are switched in the figure legend relative to the figure itself. In Supplemental Fig 2 b, black and white images were shown to support the *daf-11* mosaic analysis. It was unclear how specific sensory neurons were identified definitively in this analysis. Did the authors carry out DiI dye-filling of these animals, or use other markers to identify the neurons? Without such markers, I do not see how the cells expressing the transgene could be confidently identified.

3. In the Results section starting at line 207 I would have appreciated it if the authors had stated that they measured fat storage after 4 hours of odorant/absence of food so that I didn't have to hunt for this important detail in the Methods. Similarly, for Fig 3f I would have appreciated it if the results or figure legend had specified that changes in fat content were measured after 2 hours in the absence of food so that I wouldn't have to hunt for this in the Methods. The authors might have also mentioned e.g. in the Fig 3 legend or Methods that EtOH is the vehicle (negative control) to make this easier to understand for non-experts.

Reviewer #1:

Mutlu et al. describe that olfaction can regulate fat storage and describe a circuit from an asymmetric olfactory neuron, via an interneuron, a neuropeptide and its receptor, eventually funneling into daf-16/FoxO activation. I think the manuscript is a good fit for Nature Communication, provided that one point gets clarified by the authors. I'm not quite understanding how the authors explain that during this olfactory information processing daf-16 does NOT appear to translocate to the nucleus in the intestine. If it doesn't translocate to the nucleus, how can it fulfill its "canonical" role on controlling fat metabolism in the intestine? I'm perhaps overlooking something here and hope that the authors can clarify this matter.

Response: We truly appreciate the reviewer's positive feedback and thank him/her for recommending our manuscript for Nature Communications. We are sorry that the reviewer was confused by our explanation related to DAF-16 nuclear translocation. We did not intend to conclude that "daf-16 does NOT appear to translocate to the nucleus in the intestine", but to explain that the increased activity of DAF-16 is not caused by enhanced nuclear distribution. In wild-type worms, DAF-16 is not excluded from the nucleus and is equally distributed between the cytoplasm and the nucleus (PMID: 11381260 and 18828672). In *daf-2* mutants, DAF-16 nuclear distribution is enhanced, leading to more nuclear DAF-16 than cytoplasmic DAF-16 (PMID: 11381260 and 11747821). However, we did not observe such an enhancement when altering the olfactory pathway, and in the *daf-11* mutants with defective olfactory asymmetry, the distribution between cytoplasmic and nuclear DAF-16 is similar to that in wild type worms. Therefore, we searched for kinases that are known to directly regulate DAF-16 transcriptional activity without altering its distribution between the cytoplasm and the nucleus, and discovered SGK-1. SGK-1 can induce the transcriptional activity of DAF-16 proteins that already exist in the nucleus (PMID: 23786484). We have revised the text to avoid the confusion (lines 329-330).

Reviewer #2:

This manuscript by Mutlu et al. describes a novel genetic pathway by which the olfactory neuron AWC in *C. elegans* modulates peripheral fat metabolism through downstream endocrine signaling and not via changes in food intake. The authors start with a broad array of chemosensory mutants expected to act in the nervous system, identifying alterations in fat metabolism by the use of state-of-the-art Raman scattering microscopy in *daf-11* mutants. The authors go on to establish a role of the AWC neurons in reciprocal regulation of body fat mass, and an elaborate signaling pathway from AWC to the AIY interneuron finally to the intestine via neuropeptide flp-1 signaling, possibly engaging the NPY-receptor ortholog NPR-4. The experimentation is compelling and rigorously conducted, and the manuscript is well written and clear, and will be of broad interest to the

readership of Nature Communications after some important points are addressed.

Response: We appreciate the reviewer finding our work interesting and compelling to the broad audience of Nature Communications.

Major comments:

The manuscript alludes to but does not do justice to the myriad literature on the so called “cephalic” phase of food sensing, which is known to modulate peripheral metabolism. While this likely involves both visual perception and olfaction, a very nice paper detailed such a mechanism in Cell in 2018 (PMC6541012). Studies such as this one should be referred to in the manuscript to bring the non-expert reader up to speed on the state of the art in the literature. Interestingly in *C. elegans* which does not have an obvious visual system, the cephalic phase is likely to be dominated by the olfactory system.

Response: We thank the reviewer for his/her suggestions. We have included additional discussion and references related to the regulation of metabolism by the cephalic phase of food sensing (lines 446-451).

The analyses of fat mass could be confounded by the presence of a moderate level of constitutive dauer formation (known of course to drive high fat mass and reduced fat oxidation). Do the presence of dauers alter interpretation of the data in Figure 1 (particularly the “grind up” analyses as done in Fig 1b? To this end the data for Figure 2D ASJ rescue are reassuring. The data shown are for day 3 adults.

Response: We thank the reviewer for raising this point. The fat storage increase in the *daf-11* mutants can be detected at L4 stage, and day-1, day-2, day-3 adult stages. As the reviewer noticed, most of our experiments use adults. In these analyses, non-dauer worms were hand-picked, and fat content levels were imaged and quantified using SRS microscopy. But biochemical assays using ground adult worms may be affected by the presence of eggs. Therefore for the data shown in Figure 1b, the biochemical experiment was conducted using L4 worms to avoid the possible interference from eggs. In this experiment, worms were grown at the permissive temperature, 15°C till L3 stage, then switched to the restrictive temperature, 25°C for 8 hours, and harvested at L4 stage. Using this experimental procedure, less than 10% of the *daf-11* mutants become dauers, and we have hand-picked and removed these few dauer worms before collecting samples. Therefore, all the results are not confounded by the presence of dauer formation. We have now described this experimental procedure in detail in the revised “Biochemical Assay” section in the Methods (lines 584-587).

Are the increases in fat manifest at earlier developmental stages?

Response: The fat storage increase in the *daf-11* mutants can be detected at L4 stage when worms are switched to restrictive temperature at L3 stage (8 hours at restrictive temperature). The fat storage increase is also detected at day-1 adult stage when worms are switched to restrictive temperature at L4 stage (12 hours at restrictive temperature). Thus, the fat storage increase can be detected in both developmental and adult stages, and can be induced within 12 hours of *daf-11* inactivation.

Do the *daf-11* mutants have a fecundity defect? Fecundity is known to impact lipid stores in the worm.

Response: The *daf-11* mutants do have a smaller brood size. However, we do not think that the increased fat phenotype is caused by the defect in fecundity. 1) As discussed in the above response, the fat storage increase can be detected at L4 stage before reproduction starts. 2) *daf-16* inactivation can suppress the fat storage increase but cannot suppress the decrease in brood size. Thus, we conclude that the fecundity and the fat storage phenotypes of the *daf-11* mutants are likely regulated via distinctive mechanisms. This new result is now shown in Supplementary Figure 4d and discussed in the main text (lines 317-321).

The lipid flux analyses are excellent—but can the authors be more specific about which “lipids” they are tracking by SRS microscopy in the deuterium labeling experiments? Are we talking about all lipids into which oleate is incorporated (membrane and neutral lipids) or mainly just neutral lipids? I would guess the latter given their predominance but it would be good to have an estimate of the preponderance of triacylglycerols in the assay.

Response: As the reviewer thought, the imaging parameters were set up for neutral lipids stored in lipid droplets in all the SRS microscopy imaging results shown in this manuscript. With this experimental set-up, membrane lipids or free fatty acids do not reach a high enough local concentration to be detected by SRS. In particular, the lipid flux analyses only track neutral lipids synthesized from deuterium-labeled oleic acids and incorporated in lipid droplets. We have revised Results (line 138, 142) and Methods parts (line 578-581) to make this point clear.

Data in figure 2C and D are improperly statistically analyzed. The comparison should be by 2-way ANOVA with post-hoc testing as the experiment has two variables—strain, and presence or absence of the rescue transgenic array. The same criticism holds true for all experiments with two variables (Fig 3D, E, F, H; 4c; 5B, C, D, E; 6C, D, F, G) which should be analyzed by two-way ANOVA.

Response: We thank the reviewer for his/her suggestion on statistical analyses, and have re-analyzed these data using two-way ANOVA and found no changes in P-values. We have revised the figure legends to state that two-way ANOVA with Sidak's ad-hoc test is used for statistical analyses. We have also updated

Methods accordingly (lines 737-738).

The data on SGK-1 based activation of DAF-16 is in contrast to what is thought to be the canonical role of AGC family kinases in inhibition of FOXO activity, particularly with AKT and SGK. However, these data are in line with work on lifespan extension in *sgk-1* gain of function mutants and how this depends upon *daf-16*. The authors may want to draw attention to this work (PMC3824081).

Response: We thank the reviewer for this suggestion. The paper that the reviewer mentioned was cited in our manuscript in line 328 where we mention SGK-1 as one of the regulatory kinases that activates DAF-16.

The authors are somewhat dismissive of the *ser-2* result that fully reverses the high fat mass in the *daf-11* mutant. *Ser-2* is a putative biogenic amine receptor, others have worked on signaling of serotonin to the intestine and regulation of body fat mass. While *ser-2* is a putative tyramine receptor, the authors may want to put this work in context.

Response: We appreciate the reviewer's interests on the *ser-2* receptor. To put *ser-2* in context, we have included additional discussion and references on the previously known role of serotonin, octopamine and dopamine in regulating metabolism (lines 484-498). As noted by the reviewer, SER-2 is a putative tyramine receptor, and the importance of tyramine signaling in metabolic regulation remains unclear. The *ser-2* receptor is known to express in a subset of head neurons. Thus unlike the *npr-4* receptor expressing in the intestine, we do not think that the *ser-2* receptor acts in the peripheral fat storage tissue, but expect its involvement in the modulation of the AWC-AIY olfactory neural circuit. We believe that understanding this modulatory role of *ser-2* will be an interesting topic for future characterization, but beyond the scope of the current manuscript. In the revised manuscript, we have included additional discussion related to these points (lines 484-498).

In general, the direct connections of FLP-1 to NPR-4 are lacking, and the mechanistic connections, while possible, are missing. Further, the idea that multiple connections (based not in the least on the authors' *ser-2* result) from AIY to intestine may govern fat mass raise the possibility that NPR may be a separate sufficiency signal not necessarily engaged directly by FLP-1. Without evidence of a FLP-1 overexpression phenotype that is mitigated by loss of NPR-4 signaling (and even then the connection could be indirect), the conclusions and implications drawn from the model (Figure 7) about direct connections should be softened somewhat.

Response: We thank the reviewer for his/her suggestion to provide additional evidence in supporting the link between the FLP neuropeptide and the NPR-4 neuropeptide receptor. As the reviewer suggested, we have knocked down *npr-4* in *flp-1* overexpressing transgenic worms that show increased fat storage, and

found that *npr-4* inactivation fully suppresses the fat storage increase. This new result confirms that NPR-4 acts downstream of FLP-1, and is shown in Supplementary Figure 5b and described in the Results section (lines 381-382). In addition, neuropeptide receptor deorphanization studies have identified FLP-1 as a ligand for NPR-4 (PMID: 23267347 and 24982652), which is added to the Results section in the revised manuscript (lines 382-383).

Minor comments:

1) *C. elegans* do not have adipose tissue, and it is confusing to call intestine adipose tissue (line 109). Please rephrase to indicate that intestine is at least one of several major fat storage sites in the worm.

Response: We thank the reviewer for this comment. We changed the sentence as he/she suggested (line 114 in the revised manuscript).

2) Can the authors speculate as to why *daf-11* mutants *ks67* and *m47* show different intensity of fat mass phenotypes?

Response: *m47* is a nonsense mutation leading to deletion of the entire guanylyl cyclase catalytic domain of DAF-11, which is therefore likely a null mutation. *ks67* is a missense mutation that alters a key residue of the catalytic domain, which may reduce cGMP generation and is therefore likely a hypomorph. This information about alleles is now added to the Results section in the revised manuscript (lines 108-111).

3) It is probably an overstatement to say that food intake is not different based upon only pharyngeal pumping rates (lines 113-122). The authors may want to indicate the limitations of such an assay in truly assessing food intake.

Response: Although pharyngeal pumping is often used as a measure of food intake in *C. elegans* research, we agree with the reviewer that worms with the same pumping rate may ingest varying amounts of bacteria and have different food intake levels. On the other hand, deuterium labeled fatty acid feeding coupled SRS imaging shows that the rate of intake and incorporation of fatty acids is similar in control and *daf-11* mutant worms. To make these statements clear, we have changed “food intake rate” to “pharyngeal pumping rate” in line 119, and mentioned that worms have not changes in fatty acid intake when describing the isotope-labeling coupled SRS microscopy results (line 139-140).

Reviewer #3:

This manuscript uses a sophisticated set of genetic tools to investigate how specific odors and olfactory neurons in *C. elegans* regulate fat metabolism. Taking advantage of a substantial body of previously established knowledge

about olfaction and fat metabolism in the worm, and using an impressive set of genetic tools that the authors and others have established previously, the manuscript is able to delineate a detailed mechanism by which an odorant acts through a series of cells and signaling molecules to ultimately affect fat storage in the intestine. Overall, the work is well done, and the manuscript describes an impressive amount of work and results. The work is significant and of broad interest in that it provides a novel level of mechanistic insight into how odorants affect fat metabolism. The results are convincing and well supported by the experimental evidence presented. The statistical analysis of the data shown is appropriate.

I list below three "major criticisms", but the authors should be able to deal with these easily by revising their writing. Three additional minor points should also be easily dealt with.

Response: We appreciate the reviewer finding our work novel, significant and of broad interest to Nature Communications audience.

Major criticisms.

1. The results on the locomotion behavior of *daf-11* mutants are confusing and I was not satisfied with how they were handled. I go into this issue in detail here because the authors conclude from their locomotion analysis that *daf-11* likely does not affect fat storage by decreasing locomotion. I was not convinced by their argument. The relevant results are presented in a paragraph in the Results in lines 113-122, in Fig 1c-e, and in Supplemental Fig 2a, 2b. The confusion results from the fact that two measurements of locomotion (Fig 1c and Supplemental Fig 2a) show no change in locomotion in *daf-11*, while another measurement (Supplemental Fig 2b) shows that *daf-11* mutants spend only about 1/3 as much time as the wild type in the high-locomotion "roaming" state. The definition of roaming includes a wide range of locomotion rates, so it is conceivable that there is no contradiction between the results shown in the three analyses. It is also possible that there really is a contradiction: maybe the data set collected for Fig 1c (a small number of worms analyzed for 1 minute each) was insufficient to properly sample worms in both the roaming and dwelling states. At the very least, the authors should provide a plausible explanation of why one measure of locomotion shows a large change in *daf-11* while two other measurements do not. In interpreting their results, the authors cite a paper for the result that *che-2* mutants also had decrease in roaming, and in the supplemental Fig 1a of this manuscript, a *che-2* mutant showed no change in fat content. The reader does not know if the same mutant allele of *che-2* was used in both analyses and whether the magnitude of the roaming changes in *che-2* are comparable to those in *daf-11*. So, I found this argument unsatisfying.

Response: We understand the reviewer's concerns. For Figure 1c, the locomotion activity of worms was recorded for one minute and then tracked using Nikon Elements AR imaging software. This analysis measures the average speed of a worm during its movement that is triggered by tapping the plate before

recording, and the result indicates that when compared to wild-type controls, the *daf-11* mutants do not move faster or slower during their locomotion. We further conducted long-term recording for 40 minutes, and performed analysis to quantify the proportion of different behavioral states. The results from this analysis are shown in Supplementary Figures 2a and 2b, and indicate that 1) when compared to wild-type worms, the *daf-11* mutants show no difference in the time being resting (less than 5um/sec speed) vs. active (more than 5um/sec speed); 2) when active, the *daf-11* mutants spend more time in dwelling (speed more than 5um/sec but less than 60um/sec), rather than roaming (speed more than 60um/sec). Thus, there is no contradiction between these two figures. We have revised the manuscript to clearly state the difference between these two analyses.

These two analyses characterize different parameters of physical activities, and show that only the proportion of dwelling and roaming behaviors is altered in the *daf-11* mutants. To date, it is unclear whether an increased roaming behavior is related to increased fat accumulation. Given that the *che-2* mutant (we have used the same allele e1033) with increased roaming behavior shows no induction of fat storage, we think there might not be a direct correlation. However, as the reviewer suggested, the magnitude of roaming changes might be important for causing changes in fat storage. Maybe like different types of exercise, roaming and dwelling locomotion behaviors exhibit different demands on lipid oxidation. It could be an interesting topic in future studies to understand whether and how different behavioral states of *C. elegans* are correlated with different metabolic states. We have revised the manuscript to include additional discussion on these points (lines 124-129).

2. In Figure 4 and Figure 6, the schematics at the top of the panels depict schematically that the *daf-11(m47)* mutation results in worms having two AWCON neurons (graphically depicted as two blue triangles). This confused me, as I do not believe such an effect of *daf-11* is cited as having been established in any reference or established by any results presented earlier in the manuscript. The earlier results in this manuscript do show that *daf-11* results in fat storage defects similar to those of the *nsy-5* mutant that is established to result in the presence of two AWCON neurons, but the AWCON identity is defined by the expression or non-expression of specific molecular markers (*str-2* and *srsx-3*) that were not examined in the *daf-11* mutant. The use of the blue and orange triangles to depict AWC type should be corrected or clarified.

Response: We are sorry that the reviewer was confused by our schematic representation of the AWC olfactory neuron circuit. In the scheme, yellow triangle represents AWC^{ON} and blue triangle represents AWC^{OFF}. Mutants with two AWC^{OFF} neurons, *nsy-5* and *daf-11*, are represented with two blue triangles, whereas mutants with two AWC^{ON}, *nsy-1*, are shown with two yellow triangles. We hope this clarifies the usage of triangles to depict AWC neuron types.

Previous studies from the Bargmann lab have shown that *daf-11* mutants lack AWC asymmetry and have two AWC^{OFF} neurons (PMID: 10571181 and 20713521). In the Results section, we have referenced these studies (lines 204-206).

3. The experiments in Figure 6 establish that NPR-4 acts in the intestine to mediate downstream effects of *daf-11* and AWCON on fat storage. I agree that these data strongly infer that that NPR-4 may be a receptor for the neuropeptide FLP-1. However, the current practice among people studying *C. elegans* neuropeptides is to not conclude definitively that a GPCR is a receptor for a neuropeptide based only on such indirect genetic evidence. The additional experiment required to make such a conclusion would be to express the GPCR on heterologous cells and show that application of the peptide activates the receptor. The authors did not do that experiment. Therefore, the wording used to talk about the hypothesis that NPR-4 may be the FLP-1 receptor needs to be chosen carefully. In the Results (lines 365-366) the authors are appropriately cautious in their wording. I would appreciate it if in the Figure 6d and 6e legend, the authors also are more cautious, i.e. instead of stating that NPR-4 is the FLP-1 receptor, the statement should be that NPR-4 is the putative or apparent FLP-1 receptor.

Response: We appreciate the reviewer's suggestion and have revised the figure legend to indicate that NPR-4 is the putative FLP-1 receptor (line 1062, 1064). At the same time, we found that neuropeptide receptor deorphanization studies have identified FLP-1 as a ligand for the NPR-4 receptor (PMID: 23267347 and 24982652), which supports our genetic analysis. In addition, as Reviewer 2 suggested, we have knocked down *npr-4* in *flp-1* overexpressing transgenic worms that show increased fat storage. We found that *npr-4* inactivation fully suppresses the fat storage increase. This new result further supports that NPR-4 acts downstream of FLP-1. In the revised manuscript, we have referenced the neuropeptide receptor deorphanization studies in the Results section and included the new result in Supplementary Figure 5b (lines 381-383).

Minor points

1. The authors should search the manuscript and supplemental material for all occurrences of "*C. elegans*" and "*Caenorhabditis*" and make sure they are all italicized.

Response: Thank the reviewer's feedback. We thoroughly proofread the manuscript and italicized "*C. elegans*" and "*Caenorhabditis elegans*" in the main text, supplemental text, and references.

2. In supplemental Fig 2, panels B and C are switched in the figure legend relative to the figure itself. In Supplemental Fig 2 b, black and white images were shown to support the *daf-11* mosaic analysis. It was unclear how specific sensory neurons were identified definitively in this analysis. Did the authors carry out Dil

dye-filling of these animals, or use other markers to identify the neurons? Without such markers, I do not see how the cells expressing the transgene could be confidently identified.

Response: Thank the reviewer's feedback. We think that the reviewer was referring to Supplemental Figure 1, instead of 2. We have switched the figure legend for Supplemental Figure 1b and 1c.

For identifying the specific neurons in mosaic analysis, we stained the subset of sensory neurons using green lipophilic dye, DiO, as we were using the transgenic line *raxEx144[Pdaf-11::daf-11::sl2::RFP]* with RFP expression in neurons. We have revised the main text to state this clearly (lines 166-167).

3. In the Results section starting at line 207 I would have appreciated it if the authors had stated that they measured fat storage after 4 hours of odorant/absence of food so that I didn't have to hunt for this important detail in the Methods. Similarly, for Fig 3f I would have appreciated it if the results or figure legend had specified that changes in fat content were measured after 2 hours in the absence of food so that I wouldn't have to hunt for this in the Methods. The authors might have also mentioned e.g. in the Fig 3 legend or Methods that EtOH is the vehicle (negative control) to make this easier to understand for non-experts.

Response: We thank the reviewer, and have made the suggested textural changes in the figure legends (lines 1008-1010, 1017-1019 and 1026).

Reviewers' comments:

Reviewer #1 (Remarks to the Author):

The authors addressed my only comment and I now fully support publication as is.

Reviewer #2 (Remarks to the Author):

This revised manuscript by Mutlu et al. is an improvement on an already strong manuscript. I am in favor of publication in Nature Communications. There are still a few minor points and statistical corrections that should be addressed prior to publication:

Could the authors conduct statistical analysis e.g. Chi Squared testing on the data in Figure 2e?

Figure 4a, 4b should be analyzed by 2-way anova—two variables daf-11 and the other mutation being analyzed.

I share the concern of another reviewer that it is possible that daf-16 nuclear localization is not seen is the relative strength of the reduction in insulin signaling from the daf-2 mutation versus the daf-11 mutation? The SGK result is a nice corroborative piece, but there could still be a difference in daf-16 localization that is not picked up by the insensitive nature of the daf-16::GFP assay used.

Grammatical comments:

Line 304 "no such an effect" should be "no such effect"

Line 318 "Different to its suppressing effect on the fat storage increase" could be clarified to "Distinct from its suppressive effect on fat storage"

Line 466 "manner" should be "manners" or "a different manner"

Line 508 "a metabolic balance" should be "metabolic balance", and "which would shed light" should probably be changed to "which may shed light".

Reviewer #3 (Remarks to the Author):

I am satisfied that in their revision and their rebuttal letter, the authors have satisfactorily addressed most of my criticisms of the original manuscript. However, there is one issue that has NOT been satisfactorily addressed and that requires a further revision. This is a simple issue that can be quickly fixed by deleting a single sentence, and I think the journal editors can review this further revision without sending the manuscript back to the reviewers.

The issue is one that was brought up by both reviewer #2 (their final "Major Comment") and by me (reviewer #3, in my third and final "major criticism"). We both remarked that definitive evidence that NPR-4 is the receptor for FLP-1 neuropeptide is lacking, and that interpretations given in the manuscript should be appropriately cautious on this point. The authors have now added an experiment suggested by reviewer #2, showing that the effects of FLP-1 overexpression can be suppressed by knocking out NPR-4. This is a nice addition to the manuscript and to the evidence that NPR-4 functions downstream of FLP-1, furthering the circumstantial evidence that NPR-4 might function directly as the FLP-1 receptor. The authors have also modified their language

to use more cautious wording, now stating that NPR-4 is a "putative" FLP-1 receptor, rather than simply asserting that NPR-4 is a FLP-1 receptor. This is also a good revision. However, the problem is that the authors have added a new sentence on lines 382-383 citing references that they claim show "deorphanization studies have identified the FLP-1 neuropeptide as a candidate ligand for the NPR-4 receptor". I have read through the two cited papers carefully and do NOT see this. The papers cited are both review articles, not primary research papers, and they describe studies showing that FLP-18, not FLP-1, may encode ligands for the NPR-4 receptor. I know of no published work asserting that FLP-1 peptides can bind and activate the NPR-4 receptor. So, the statement added on lines 382-383 is not accurate and must be deleted. I note that the indirect genetic evidence shown in this manuscript that FLP-1 function depends on NPR-4 is very good and stands on its own. We just need to be clear about the fact that nobody has ever shown directly that FLP-1 peptides can bind and activate the NPR-4 receptor - this is purely an inference from indirect genetic evidence.

Reviewer #1:

The authors addressed my only comment and I now fully support publication as is.

Reviewer #2:

This revised manuscript by Mutlu et al. is an improvement on an already strong manuscript. I am in favor of publication in Nature Communications. There are still a few minor points and statistical corrections that should be addressed prior to publication:

Could the authors conduct statistical analysis e.g. Chi Squared testing on the data in Figure 2e?

Response: Based on the reviewer's suggestion, we have conducted statistical analysis on the data in Figure 2e. We used Fisher's exact testing because Chi-square cannot be used when the entire row is "0" which is the case when comparing wild type worms with AWC ablation mutants, *nsy-5* and *nsy-1*. To present the statistical analysis data, we have now included two more columns on the table in Figure 2e, with *P* values comparing the test groups with wild type worms and *daf-11* mutants, respectively.

These statistical analyses further support the interpretation of the results: specific restoration of *daf-11* expression in AWC neurons has no effect on dauer formation in *daf-11* mutants, whereas its restoration in ASJ neurons rescues the Daf-c phenotype. We have also conducted Fisher's exact test to further support the results shown in Supplementary Figure 4c.

Figure 4a, 4b should be analyzed by 2-way anova—two variables *daf-11* and the other mutation being analyzed.

Response: We have re-analyzed the data in Figure 4a and Figure 4b by two-way ANOVA. The statistical significance is unchanged. We have revised the figure legend to be consistent with this change..

I share the concern of another reviewer that it is possible that *daf-16* nuclear localization is not seen is the relative strength of the reduction in insulin signaling from the *daf-2* mutation versus the *daf-11* mutation? The SGK result is a nice corroborative piece, but there could still be a difference in *daf-16* localization that is not picked up by the insensitive nature of the *daf-16::GFP* assay used.

Response: We thank the reviewer for these comments. We found the additive effect from the *daf-2; daf-11* double mutants on fat storage increase and the differences in DAF-16 nuclear localization between the *daf-2* and the *daf-11* mutants. These results suggest that *daf-2* and *daf-11* may act in parallel pathways to regulate fat storage. However, we agree with the reviewer that there is still the possibility where *daf-11* and *daf-2* are in the same or overlapping pathway, and the *daf-11* inactivation enhances fat storage increase in the *daf-2* mutant by further reducing insulin signaling. We revised the main text to clearly state these two possibilities (lines 288-292).

Grammatical comments:

Line 304 "no such an effect" should be "no such effect"

Line 318 “Different to its suppressing effect on the fat storage increase” could be clarified to “Distinct from its suppressive effect on fat storage”

Line 466 “manner” should be “manners” or “a different manner”

Line 508 “a metabolic balance” should be “metabolic balance”, and “which would shed light” should probably be changed to “which may shed light”.

Response: We thank the reviewer for carefully reading our manuscript. We have corrected these grammar errors based on his/her suggestions.

Reviewer #3:

I am satisfied that in their revision and their rebuttal letter, the authors have satisfactorily addressed most of my criticisms of the original manuscript. However, there is one issue that has NOT been satisfactorily addressed and that requires a further revision. This is a simple issue that can be quickly fixed by deleting a single sentence, and I think the journal editors can review this further revision without sending the manuscript back to the reviewers.

The issue is one that was brought up by both reviewer #2 (their final "Major Comment") and by me (reviewer #3, in my third and final "major criticism"). We both remarked that definitive evidence that NPR-4 is the receptor for FLP-1 neuropeptide is lacking, and that interpretations given in the manuscript should be appropriately cautious on this point. The authors have now added an experiment suggested by reviewer #2, showing that the effects of FLP-1 overexpression can be suppressed by knocking out NPR-4. This is a nice addition to the manuscript and to the evidence that NPR-4 functions downstream of FLP-1, furthering the circumstantial evidence that NPR-4 might function directly as the FLP-1 receptor. The authors have also modified their language to use more cautious wording, now stating that NPR-4 is a "putative" FLP-1 receptor, rather than simply asserting that NPR-4 is a FLP-1 receptor. This is also a good revision. However, the problem is that the authors have added a new sentence on lines 382-383 citing references that they claim show "deorphanization studies have identified the FLP-1 neuropeptide as a candidate ligand for the NPR-4 receptor". I have read through the two cited papers carefully and do NOT see this. The papers cited are both review articles, not primary research papers, and they describe studies showing that FLP-18, not FLP-1, may encode ligands for the NPR-4 receptor. I know of no published work asserting that FLP-1 peptides can bind and activate the NPR-4 receptor. So, the statement added on lines 382-383 is not accurate and must be deleted. I note that the indirect genetic evidence shown in this manuscript that FLP-1 function depends on NPR-4 is very good and stands on its own. We just need to be clear about the fact that nobody has ever shown directly that FLP-1 peptides can bind and activate the NPR-4 receptor - this is purely an inference from indirect genetic evidence.

Response: We thank the reviewer for his/her suggestions that improved our manuscript. As suggested by the reviewer, we deleted the sentence on lines 382-383.

REVIEWERS' COMMENTS:

Reviewer #2 (Remarks to the Author):

I am perfectly satisfied with the revision which is well matched for publication in Nature Communications.